# Mechanisms of feedback inhibition and sequential firing of active sites in plant aspartate transcarbamoylase

Leo Bellin [1], Francisco Del Caño-Ochoa [2], Adrián Velázquez-Campoy [3,4,5,6,7], Torsten Möhlmann [1✉] & Santiago Ramón-Maiques [2,8✉]

Aspartate transcarbamoylase (ATC), an essential enzyme for de novo pyrimidine biosynthesis, is uniquely regulated in plants by feedback inhibition of uridine 5-monophosphate (UMP). Despite its importance in plant growth, the structure of this UMP-controlled ATC and the regulatory mechanism remain unknown. Here, we report the crystal structures of Arabidopsis ATC trimer free and bound to UMP, complexed to a transition-state analog or bearing a mutation that turns the enzyme insensitive to UMP. We found that UMP binds and blocks the ATC active site, directly competing with the binding of the substrates. We also prove that UMP recognition relies on a loop exclusively conserved in plants that is also responsible for the sequential firing of the active sites. In this work, we describe unique regulatory and catalytic properties of plant ATCs that could be exploited to modulate de novo pyrimidine synthesis and plant growth.

[1] Pflanzenphysiologie, Fachbereich Biologie, Universität Kaiserslautern, Erwin-Schrödinger-Strasse, D-67663 Kaiserslautern, Germany. [2] Instituto de Biomedicina de Valencia (IBV-CSIC), Jaime Roig 11, 46010 Valencia, Spain. [3] Institute for Biocomputation and Physics of Complex Systems (BIFI), Joint Units IQFR-CSIC-BIFI, and GBsC-CSIC-BIFI, Universidad de Zaragoza, 50018 Zaragoza, Spain. [4] Departamento de Bioquímica y Biología Molecular y Celular, Universidad de Zaragoza, 50009 Zaragoza, Spain. [5] Instituto de Investigación Sanitaria de Aragón (IIS Aragón), 50009 Zaragoza, Spain. [6] Centro de Investigación Biomédica en Red en el Área Temática de Enfermedades Hepáticas Digestivas (CIBERehd), 28029 Madrid, Spain. [7] Fundación ARAID, Gobierno de Aragón, 50018 Zaragoza, Spain. [8] Group 739, Centro de Investigación Biomédica en Red de Enfermedades Raras (CIBERER)- Instituto de Salud Carlos III, Valencia, Spain. ✉email: moehlmann@biologie.uni-kl.de; sramon@ibv.csic.es

**P**yrimidine nucleotides are crucial to all living organisms as components of nucleic acids as well as cofactors in the synthesis of sugars, polysaccharides, glycoproteins, and phospholipids[1,2]. However, much remains unknown in plants about the unique organization, regulation, and localization of the enzymes required for de novo biosynthesis of pyrimidines[1], which are potential targets for crop improvement and weed control. This metabolic pathway starts in the chloroplast, where aspartate transcarbamoylase (ATC) catalyzes the condensation of carbamoyl aspartate from aspartate (Asp) and carbamoyl phosphate (CP)[1,3] (Fig. 1a). ATC can be anchored to the inner plastid

membrane[4], which might facilitate the channeling of carbamoyl aspartate to a cytosolic dihydroorotase (DHO) potentially associated to the outer plastid membrane[1]. The dihydroorotate produced by DHO diffuses to the mitochondrial intermembrane space and is oxidized to orotate by dihydroorotate dehydrogenase (DHODH), a membrane flavoprotein coupled to the respiratory chain[3,5]. Orotate returns to the cytosol and is transformed to uridine 5-monophosphate (UMP), the precursor of all pyrimidine nucleotides, in a two-step reaction catalyzed by UMP synthetase (UMPS). Importantly, only in plants, UMP inhibits the activity of ATC by a yet unknown mechanism, creating a feedback loop that

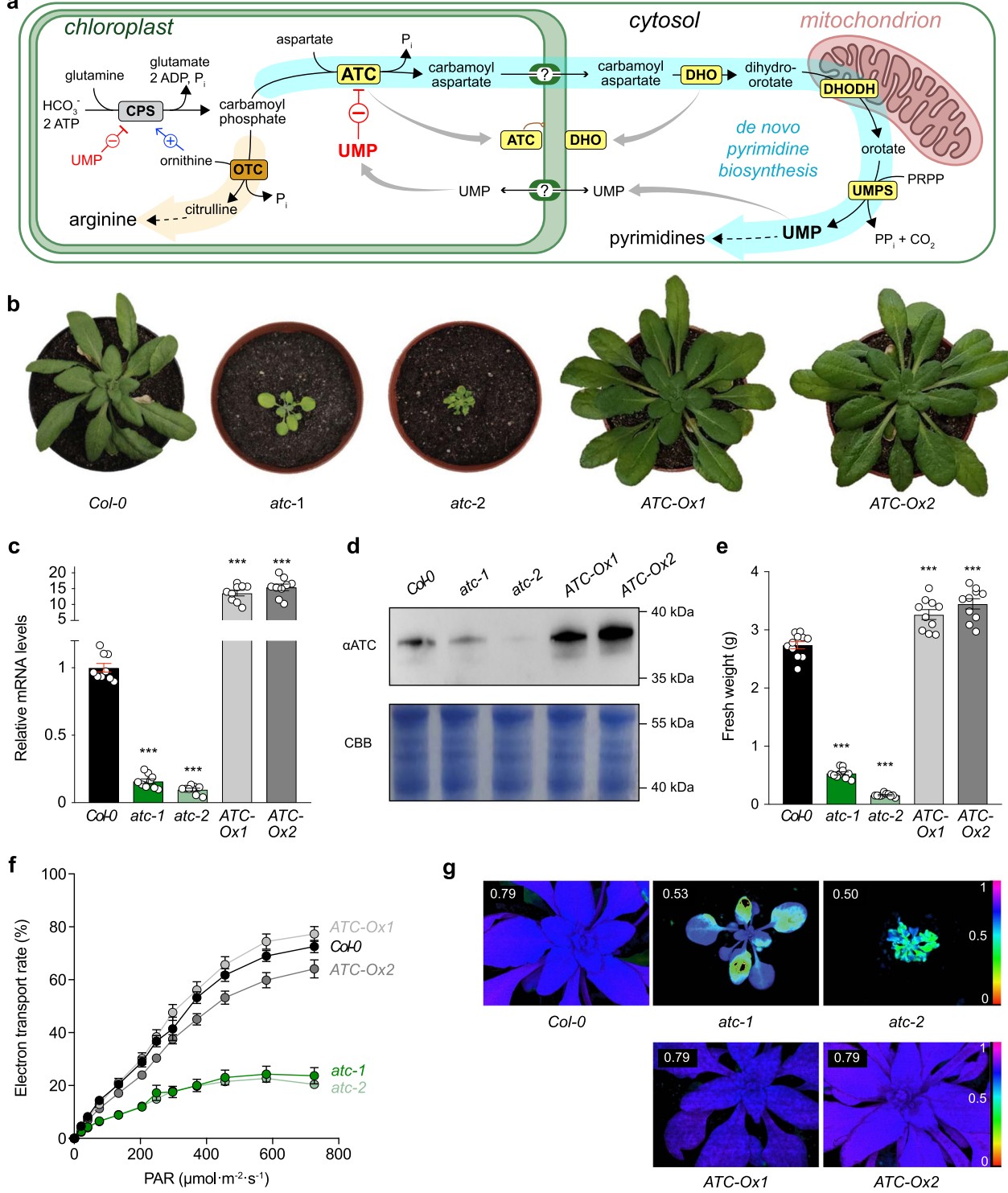

**Fig. 1 ATC central activity in de novo pyrimidine synthesis and plant growth. a** Scheme of de novo pyrimidine biosynthesis pathway (highlighted in cyan with enzymatic activities in yellow background) and arginine synthesis (in orange) in plants. ATC, aspartate transcarbamoylase; DHO, dihydroorotase; DHODH, dihydroorotate dehydrogenase; UMPS, UMP synthetase; CPS, carbamoyl phosphate synthetase; OTC, ornithine transcarbamoylase. Allosteric inhibition by UMP and activation by ornithine are indicated by red and blue lines, respectively. **b** Arabidopsis Col-0 and ATC downregulated (atc-1 and -2) or overexpressing lines (ATC-Ox1 and 2) grown for 4 weeks in a 14 h light and 10 h dark regime. **c** ATC transcript levels in knockdown and overexpressing lines relative to Col-0 (n = 9). ATC transcript levels relative to actin: $9.5 \times 10^{-3}$ (Col-0), $1.5 \times 10^{-3}$ (atc-1), $0.93 \times 10^{-4}$ (atc-2), 0.13 (ATC-Ox1), and 0.15 (and ATC-Ox2). p-values are $5.722 \times 10^{-14}$ for atc-1, $5.93 \times 10^{-15}$ for atc-2, $1.056 \times 10^{-10}$ for ATC-Ox1, and $1.016 \times 10^{-10}$ for ATC-Ox2. **d** Immunoblot with anti-ATC antibody on whole leaf extracts; Coomassie Brilliant Blue (CBB) stained SDS-PAGE was used as loading control. The ATC protein levels relative to Col-0 quantified from n-different experiments are: atc-1, 0.34 ± 0.17 (n = 5); atc-2, 0.05 ± 0.024 (n = 5); ATC-Ox1, 2.9 ± 0.44 (n = 4); and ATC-Ox2, 2.5 ± 0.87 (n = 4). **e** Fresh weight quantification (n = 10). **f** Electron transport rate (ETR) determined by PAM in standard light curve setting (n = 12 for Col-0, n = 8 for atc-1, n = 11 for atc-2, n = 12 for ATC-Ox1, n = 11 for ATC-Ox2). p-values are $2.879 \times 10^{-18}$ for atc-1, $4.563 \times 10^{-20}$ for atc-2, $3.692 \times 10^{-5}$ for ATC-Ox1, $1.142 \times 10^{-6}$ for ATC-Ox2. **g** False color presentation of maximal photosynthesis yield monitored by PAM of lines shown in **f**. Error bars indicate standard error of the mean. Asterisks depict significant changes between the different lines referring to Col-0 control according to one-way ANOVA followed by Tukey's multiple comparison test ($^*p < 0.05$, $^{**}p < 0.01$, $^{***}p < 0.001$).

controls the flux through the pathway[6–9] (Fig. 1a). UMP also inhibits carbamoyl phosphate synthetase (CPS), the chloroplast enzyme producing CP for both the synthesis of pyrimidines and arginine[10], although this effect is counteracted by ornithine to sustain the production of arginine[11,12] (Fig. 1a). Thus, ATC is the major regulated enzyme for de novo synthesis of UMP in plants[13], but yet the feedback mechanism that makes this enzyme different from any other ATC remains uncharacterized.

In general, ATCs consist of a catalytic homotrimer with three active sites in between the subunits that can be allosterically regulated by association with other proteins. The *Escherichia coli* ATC, for instance, is formed by association of two catalytic trimers with three dimers of regulatory subunits[14] responsible for the binding of nucleotides that diminish (UTP and CTP) or enhance (ATP) the activity[15,16]. Other prokaryotic ATCs lack regulatory subunits and thus, are insensitive to nucleotides[17,18]. In eukaryotes other than plants, ATC is fused together with CPS into a single multienzymatic protein named CAD that also contains an active DHO (animals) or an inactive DHO-like domain (fungi)[10,19–21]. The ATC domain of CAD also forms homotrimers, favoring the assembly of the protein into large hexameric particles[22–24], and its activity is modulated by the binding of UTP to an allosteric region within CPS[25,26]. In striking contrast, plant ATCs consist of a UMP-inhibitable homotrimer with no associated subunits, meaning that both the catalytic and regulatory sites must reside within the same polypeptide chain[27]. However, despite the wealth of biochemical and structural knowledge on ATCs from prokaryotes, fungi, and animals, there is no structural information of any plant ATC so far. Thus, the putative binding site for UMP and the catalytic and regulatory mechanisms of ATC in plants remain unknown.

Here we show that the development of *Arabidopsis thaliana* (Arabidopsis) can be severely impaired or enhanced by the expression level of ATC. To understand this fundamental activity, we determined the crystal structure of Arabidopsis ATC free and bound to UMP, in complex with a transition-state analog, with CP or bearing a site-specific mutation that turns the enzyme insensitive to UMP. The structural, mutagenesis, and biochemical analyses reveal unique catalytic and regulatory properties of plant ATCs, suggesting new strategies to control de novo pyrimidine synthesis and plant growth.

## Results

**ATC is key for plant growth and efficient photosynthesis.** To explore the importance of ATC for plant growth we used artificial microRNA (amiRNA) to knockdown ATC (At3g20330) in Arabidopsis. Two selected lines, atc-1 and atc-2, exhibited 16% and 10% residual ATC transcript and a 3-fold or 20-fold drop in protein levels compared to wild-type (WT; Col-0) controls (Fig. 1b–d). Conversely,

we constitutively overexpressed ATC in two Arabidopsis lines, ATC-Ox1 and ATC-Ox2, which showed 13- and 16-fold increase in ATC transcript and a 2.9-fold increase in protein levels (Fig. 1b–d). After 4 weeks on soil, atc-1 and atc-2 downregulated lines showed a strong reduction of growth, with fresh weights of 19% (0.53 ± 0.09 g plant$^{-1}$) and 6% (0.16 ± 0.03 g plant$^{-1}$) of the Col-0 control plants (2.73 ± 0.21 g plant$^{-1}$) (Fig. 1b, e). In contrast, ATC-Ox1 and ATC-Ox2 showed increased growth with fresh weights of 119% (3.26 ± 0.09 g plant$^{-1}$) and 126% (3.45 ± 0.09 g plant$^{-1}$) compared to Col-0, respectively (Fig. 1b, e).

ATC downregulated lines also showed pale leaves, suggesting lower chlorophyll levels, and presumably impaired photosynthesis, whereas ATC-Ox lines exhibited no phenotypic alterations other than the bigger size (Fig. 1b). Because of the pale leaf coloration, 4 week old atc plants were subjected to pulse-amplitude-modulation (PAM) fluorometry, which measures chlorophyll fluorescence as an indicator of the photosynthetic capacity. The electron transport rate (ETR) of atc lines was less than 20% of Col-0, whereas ATC-Ox lines showed no significant differences (Fig. 1f). The pronounced ETR decrease was in line with a reduction in the maximal photosynthetic efficiency (Fv/Fm), with values of 0.53 ± 0.03 and 0.50 ± 0.02 for atc-1 and -2, respectively, which are markedly lower than the 0.79 ± 0.01 measured in controls (Fig. 1g). Again, Fv/Fm values in ATC-Ox lines were similar to Col-0.

These results, together with previous studies[13,28], demonstrate a regulatory role of pyrimidine de novo synthesis in plant growth and a key function of ATC herein.

**Crystal structure of Arabidopsis ATC bound to UMP.** To investigate this central enzymatic activity, we attempted to produce the ATC from Arabidopsis, a 390 amino acid (aa) precursor protein with an N-terminal chloroplast transit peptide (Fig. 2a and Supplementary Fig. 1). Having difficulties to express the full-length protein in *E. coli*, we tested different N-terminal truncated forms. One construct spanning aa 82–390 (named atATC) was purified as a stable homotrimer (Fig. 2b), and matched in size (excluding the 2.6 kDa fusion tag) the 36 kDa mature enzyme in whole leaf extracts (Fig. 2c). atATC produced diffraction quality crystals readily and the structure was determined at 1.7 Å resolution (Supplementary Table 1 and Supplementary Fig. 2). The structure of the atATC trimer resembles a three-bladed propeller with a concave face holding three active sites in between subunits, thus preserving the overall architecture of the transcarbamoylase family[29] (Fig. 2d). Each subunit folds into two subdomains of similar size: an N-domain (aa 82–221 and 374–390) occupying the center of the trimer and holding the binding site for CP, and a C-domain (aa 222–373) bearing the Asp binding site (Fig. 2a, d and Supplementary Figs. 1 and 3). The relative orientation of the

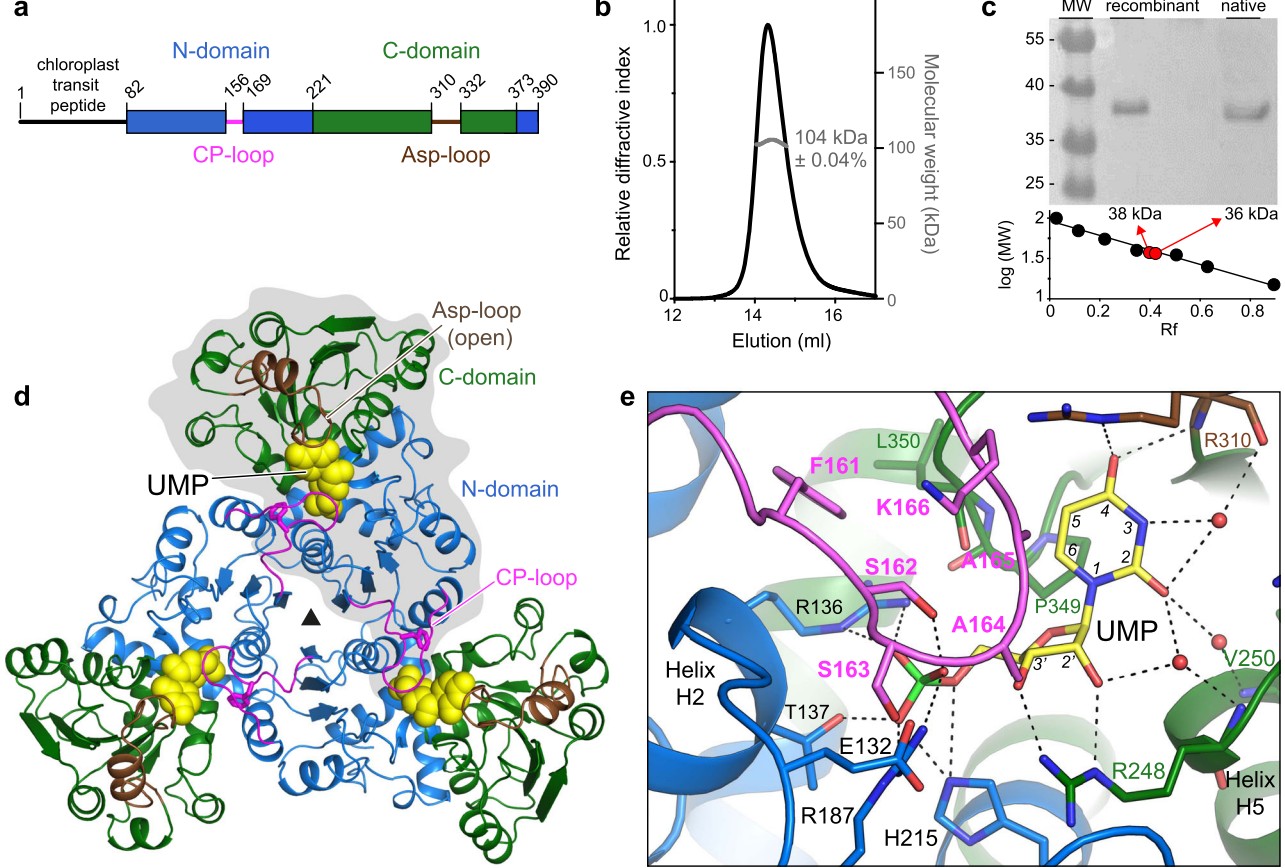

**Fig. 2 Structure of Arabidopsis ATC bound to UMP. a** Scheme of Arabidopsis ATC protein. **b** SEC-MALS analysis of purified atATC proves the formation of a homotrimer. **c** Immunoblot of atATC and mature ATC from leaf extract. The experiment was performed three times with identical results. **d** Crystal structure of atATC trimer with each subunit bound to one molecule of UMP (shown as yellow spheres). One subunit is shown on gray background. Protein domains are colored as in (**a**). **e** Detail of the active site with UMP bound. Water molecules are shown as red spheres. Electrostatic interactions are indicated as dashed lines.

domains is similar to the open conformation observed in other ATCs crystallized without ligands (Supplementary Fig. 3)[23,24,30]. In addition, atATC has the CP-loop (aa 156–169) and Asp-loop (aa 309–332) (Fig. 2a, d) that, as in other ATCs[16,24], undergo large conformational movements upon substrate binding (see below).

Unexpectedly, additional electron density in each active site indicated the presence of a molecule of UMP captured during protein expression and kept throughout the purification and crystallization process (Fig. 2d, e and Supplementary Fig. 2). The nucleotide fills the active site, with the ribose in C3′ endo pucker and the base in *anti* conformation (Fig. 2e). The phosphate binds near the N-end of helix H2 and interacts with R136, T137, R187, and H215 (at the N-domain), whereas the ribose 2′- and 3′-OH bind to the side chain of R248 (C-domain). The 4-O atom of the pyrimidine ring interacts with R310 (Asp-loop), and the 2-O and 3-NH bind through three waters to R310, R248, and V250 (C-domain), whereas the C5 and C6 atoms make Van der Waals contacts with the [349]PLP[351] loop (C-domain). In addition, the CP-loop from the adjacent subunit interacts with the inhibitor through Van der Waals contacts of residues A164 and A165 and makes a H-bond between S162 and the phosphate (Fig. 2d, e).

Next, we freed the enzyme of UMP by a gel filtration procedure (Supplementary Fig. 4) and determined the crystal structure of the apo form at 3.1 Å resolution (Supplementary Table 1). The structure turned to be similar to the UMP-inhibited conformation except for aa 160–166 of the CP-loop that appear disordered in the absence of the nucleotide (Supplementary Fig. 3).

**atATC only binds one molecule of PALA per trimer**. We investigated the effect of PALA [N-(phosphonacetyl)-L-aspartate], a potent ATC inhibitor with structural features of both substrates that mimics the transition-state of the reaction[31–33]. Seedling assays in the presence of 0.2 mM or 0.4 mM PALA showed a decrease in fresh weight to 59% or 23%, respectively, compared to untreated seedlings (3 times 10 seedlings were weighted per treatment, $n = 3$). Root length in untreated seedlings was $2.68 \pm 0.49$ cm and was reduced to 36% at 0.2 mM PALA ($0.96 \pm 0.22$ cm) and to 8% at 0.4 mM PALA ($0.21 \pm 0.09$ cm) ($n = 30$) (Fig. 3a). These results support the reduced growth observed in *atc* downregulated lines (Fig. 1b–d) and agree with previous PALA-inhibition studies[32]. Chlorosis was also apparent in PALA-treated seedlings (Fig. 3a), further endorsing the effect of reduced ATC levels on chloroplast functionality (Fig. 1f, g).

To gain further insight into the reaction mechanism, we determined the structure of atATC in complex with the transition-state analog at 1.6 Å resolution (Supplementary Table 1). Surprisingly, the structure showed the atATC trimer with PALA bound to only one of the subunits (Fig. 3b and Supplementary Fig. 5). This subunit undergoes a 10° hinge closure of the N- and C-domains and a 24° rigid body rotation of the Asp-loop (Fig. 3c), emulating the movement needed to bring CP and Asp in close contact to favor the reaction[33,34]. In contrast, the other two subunits exhibit an open conformation, similar to the apo or UMP-bound states, and have the active sites empty or with two sulfate ions and one glycerol molecule from the crystallization solution (Fig. 3b and Supplementary Fig. 5). Only

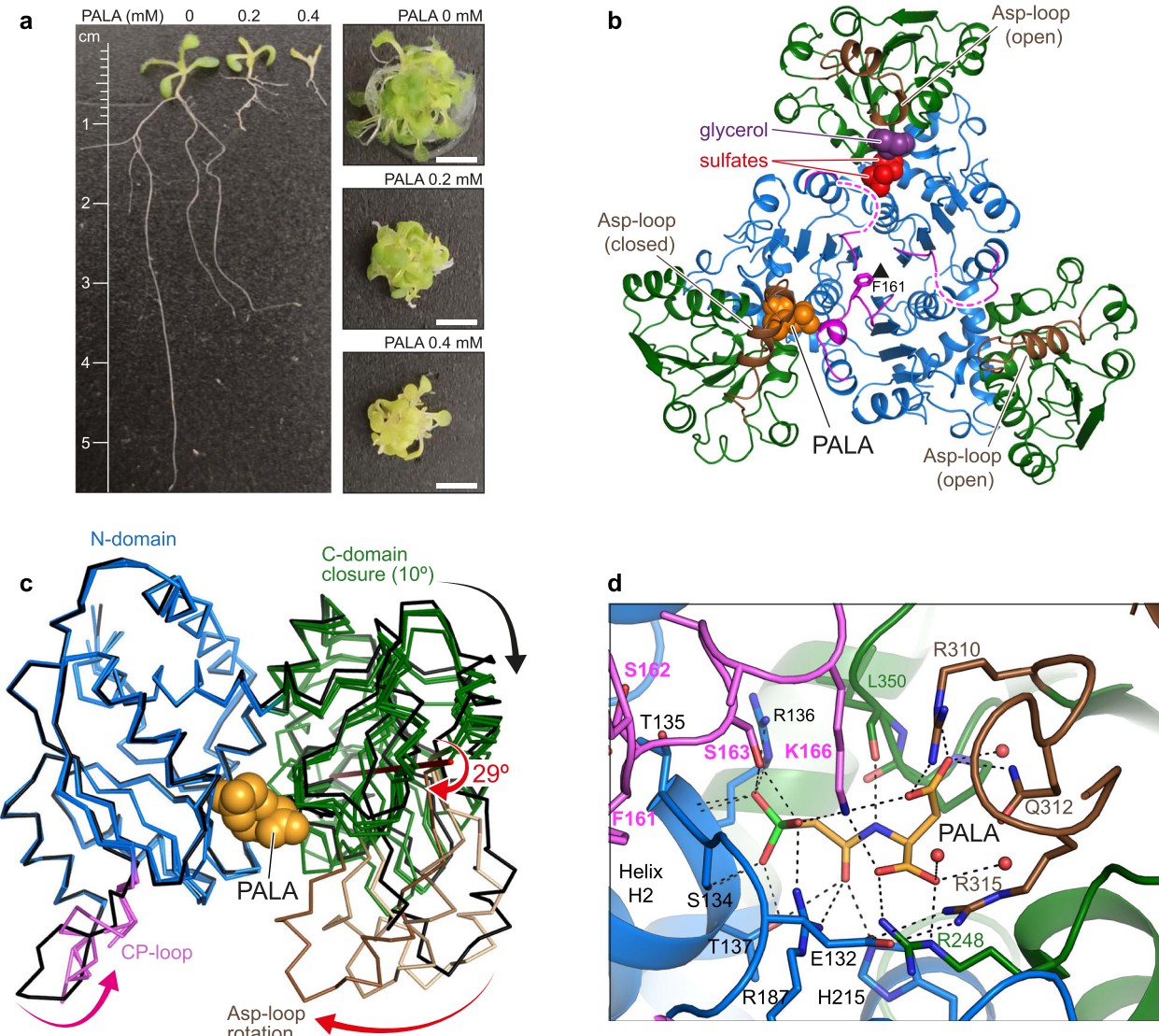

**Fig. 3 Inhibition of atATC by the transition-state analog PALA. a** Inhibition of Arabidopsis seedlings after 7 days treatment with PALA. Images are typical examples of 30 seedlings viewed per treatment. Scale bars indicate 0.5 cm. **b** Crystal structure of atATC trimer in complex with only one molecule of PALA (shown as orange spheres). Dashed lines indicate flexibly disordered CP-loops. **c** Superposition of the three subunits in the PALA-bound trimer (colored as in **b**) and in the UMP-bound subunit (in black). The arrows indicate the closure of the subunit upon PALA binding. The Asp-loop undergoes a 29° rotation around an axis that has been represented and colored in red. **d** Detail of the binding of PALA to the active site. Water molecules are shown as red spheres. Electrostatic interactions are indicated as dashed lines.

the CP-loop interacting with PALA is well-defined in the electron density map, whereas the other CP-loops are flexibly disordered (Fig. 3b).

The substoichiometric binding of PALA is remarkable, since other ATCs bound three molecules of PALA per trimer[24,34–37] and the interactions with the transition-state analog are virtually identical to those observed in atATC (Fig. 3d and Supplementary Fig. 6). The phosphonate group of PALA binds to the N-end of helix H2 (N-domain) and the O atom of the carbamate moiety interacts with T137, R187, and H215 (N-domain), whereas the N atom binds to L350 (C-domain). Also, the α-carboxylate group binds to R248 (C-domain) and the β-carboxylate binds to R310 and Q312 (Asp-loop). In addition, the CP-loop from the adjacent subunit binds through S163 to the phosphonate moiety, and places K166 at interacting distance of the phosphonate and the α- and β-carboxylates (Fig. 3d).

Isothermal titration calorimetry (ITC) analyses confirmed that PALA binds to only one site per trimer ($K_D^{PALA} = 0.6\ \mu M$) and

somehow blocks the entrance of subsequent PALA molecules to the other sites (Supplementary Table 2 and Supplementary Fig. 7). This negative cooperativity effect is specific for PALA, since the unoccupied subunits can still bind CP ($K_D^{CP} = 140\ \mu M$) or UMP ($K_D^{UMP} = 1.2\ \mu M$). We also observed negative cooperativity, but to a lesser extent, in the titration with UMP, since the nucleotide binds with a $K_D^{UMP} = 0.2\ \mu M$ to the first site and reduces 10-fold the affinity of the other subunits. In turn, CP showed equal affinity for the three active sites ($K_D^{CP} = 77\ \mu M$).

These results strongly suggested that despite the overall structural similarity with other ATCs, the atATC trimer uses a mechanism of communication between active sites that allows only one subunit to attain the closed catalytic conformation.

**The CP-loop blocks the simultaneous closure of the subunits.** The explanation for the unusual binding of PALA to atATC was

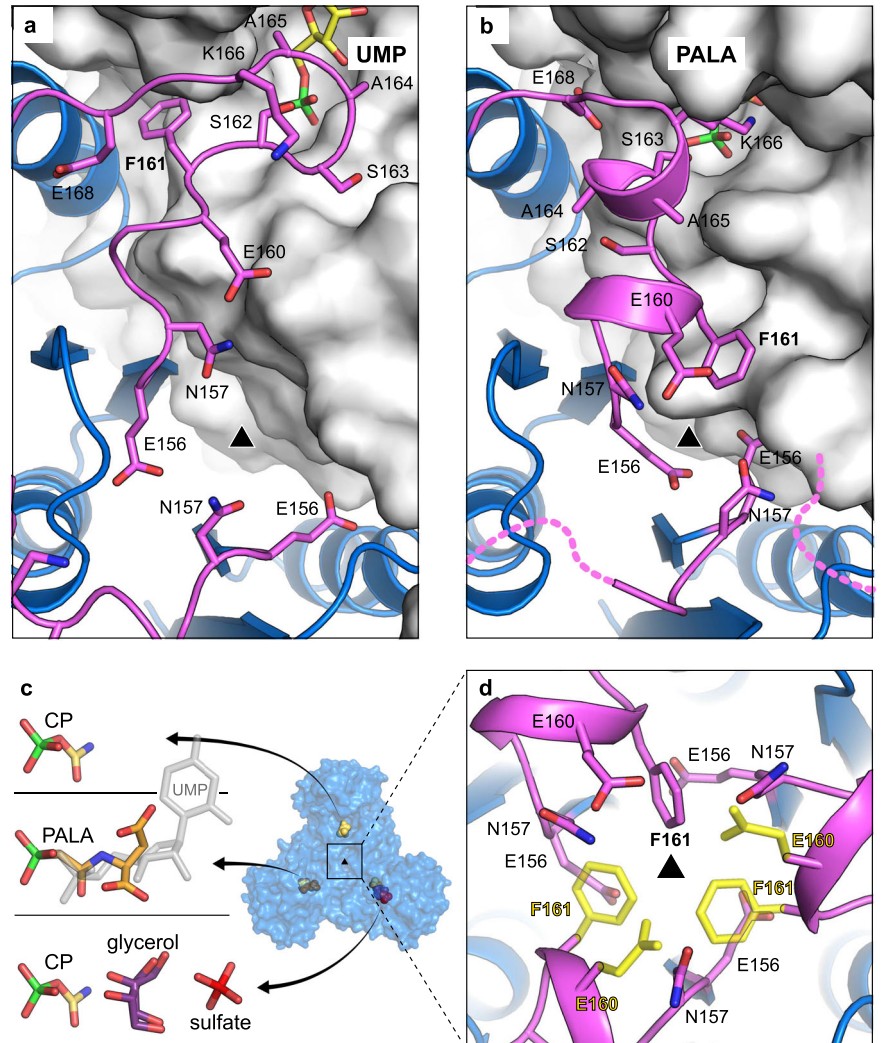

**Fig. 4 The CP-loop folds in two different conformations for UMP or PALA binding. a**, **b** Cartoon representation of the CP-loop (in magenta) over the active site of the adjacent subunit (shown in surface representation) bound to UMP (**a**) or PALA (**b**). **c** Structure of atATC with one subunit bound to PALA, a second one with CP, and the third having CP, glycerol, and one sulfate ion. A UMP molecule is shown in semitransparent gray to compare the position relative to the ligands. **d** Detail view along the three-fold axis. The side chains of E160 and F161 are only seen in one subunit. Modeling of these two residues in the other two subunits (shown in yellow semitransparent representation) causes steric clash and charge repulsion.

likely at the CP-loop, as the most distinct element compared to other non-plant ATCs (Fig. 2a and Supplementary Fig. 1). This loop is flexible in the absence of ligands (Fig. 3b and Supplementary Fig. 3) but adopts two distinct conformations whether UMP or PALA are bound to the adjacent subunit (Figs. 2e and 3d). With UMP, the CP-loop folds in an extended "inhibited" conformation, with A164, A165 and S162 interacting with the nucleotide, S163 and K166 pointing outwards the active site, and the side chain of F161 inserted in between subunits (Fig. 4a). In turn, upon PALA binding, the CP-loop rearranges into two short and nearly perpendicular $3_{10}$ α-helices, placing S163 and K166 to interact with the transition-state analog and moving A164, A165, and S162 outwards the active site (Fig. 4b). In this "active" conformation, F161 flips 180° compared to the position with UMP, and projects towards the trimer three-fold axis, where intersubunit distances are shortened by the interactions between neighbor E156 residues (Fig. 4b). These tight contacts at the center of the trimer are not observed in other ATCs bound to PALA (Supplementary Fig. 6), suggesting that the position of F161 may prevent other CP-loops from reaching a similar active conformation.

Two additional atATC structures reinforced this hypothesis. One structure, obtained from crystals with PALA and soaked in CP (Supplementary Table 1), showed a trimer with one subunit bound to PALA, a second subunit with CP, and a third subunit with CP and with one glycerol and one sulfate ion filling the Asp binding site (Fig. 4c and Supplementary Fig. 5). The second structure, obtained by co-crystallization with CP, showed all three subunits in the trimer bound to CP (Supplementary Table 1 and Supplementary Fig. 5). In both structures, the three CP-loops in the trimer fold in an active conformation but show poor electron density compared to the rest of the protein. In fact, E160 and F161 were traced in only one subunit, and modeling in similar conformation in the other subunits caused steric clash and charge repulsion (Fig. 4d).

**Mutant F161A is not inhibited by UMP and binds three PALAs.** To further test the role of the CP-loop, we replaced F161 with Ala (F161A). The mutation did not affect the solubility nor the oligomeric state of the protein, but changed the susceptibility of the enzyme to UMP, PALA, or to high

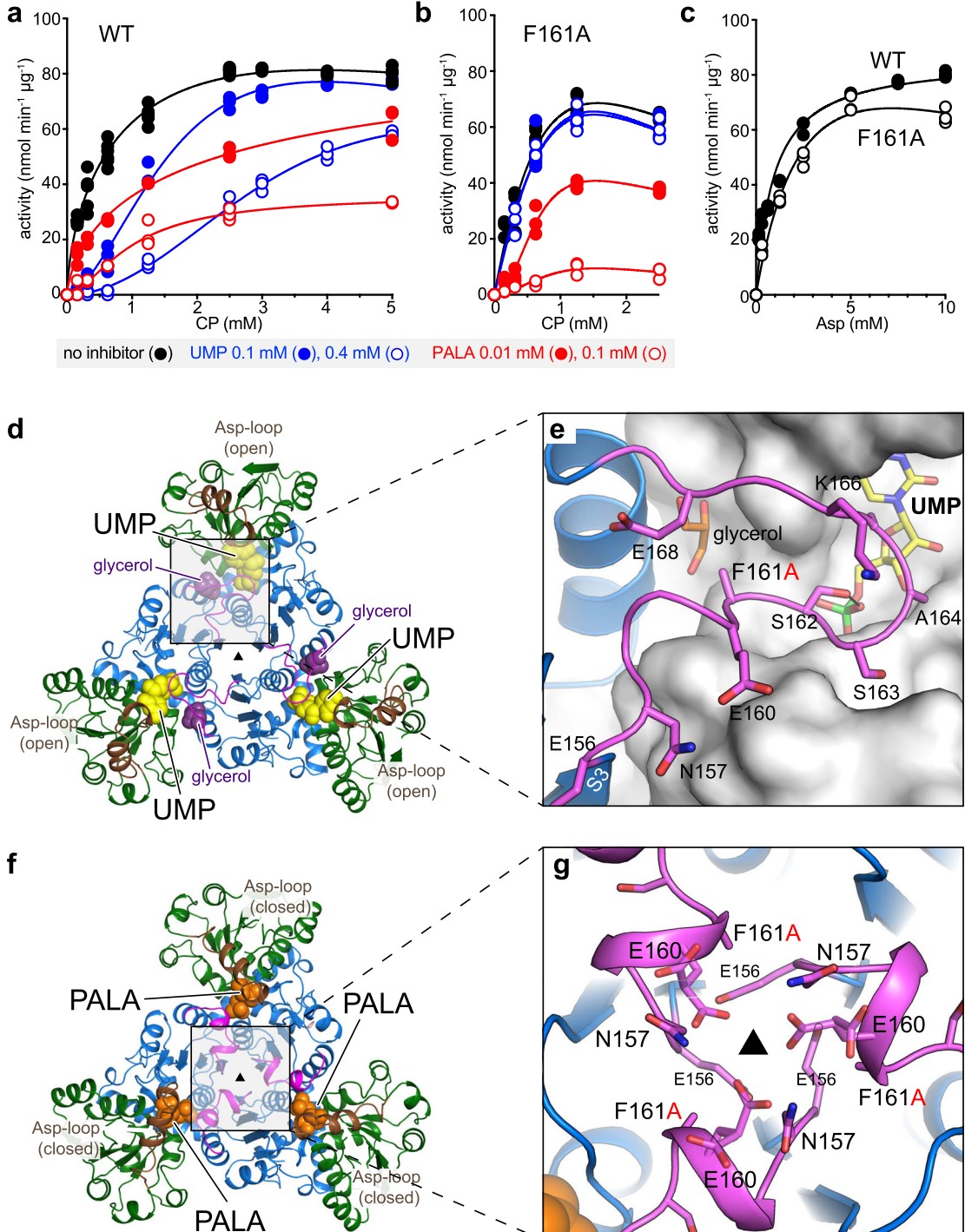

**Fig. 5 Activity and crystal structure of atATC mutant F161A. a, b** CP saturation curves of atATC WT (**a**) and F161A (**b**) in the absence and presence of UMP or PALA. **c** Asp saturation curves for WT and F161A. Equations to fit the kinetic data are detailed in Methods. **d, e** Crystal structure of atATC-F161A trimer complexed with UMP (**d**), and detail of the interactions of the CP-loop (**e**), showing a glycerol molecule replacing the missing F161 side chain. **f, g** Crystal structure of F161A with PALA bound to the three active sites (**f**), and detail of the interactions around the molecular axis (**g**).

concentrations of the substrates. Initial-rate plots of WT with CP as variable ligand are hyperbolic in the absence of UMP ($V_{max} = 93.23 \pm 2.53$ nmol min$^{-1}$ µg$^{-1}$, $K_{0.5}^{CP} = 0.46 \pm 0.04$ mM, $K_{0.5}^{Asp} = 0.94 \pm 0.10$ mM), but turn sigmoidal in the presence of UMP, with a Hill-coefficient $h = 2.2$, indicating positive cooperativity for CP binding (Fig. 5a and Supplementary Fig. 8), as previously described for wheat-germ ATC[7].

In contrast, parallel assays with F161A proved that although the catalytic activity is highly similar to the WT ($V_{max} = 115.0 \pm 14.62$ nmol min$^{-1}$ µg$^{-1}$, $K_{0.5}^{CP} = 0.62 \pm 0.13$ mM, $K_{0.5}^{Asp} = 3.20 \pm 1.03$ mM), the enzyme is not inhibited by UMP and also becomes more sensitive to the presence of PALA (Fig. 5a and Supplementary Fig. 8). In addition, F161A showed decreased activity at high substrate concentrations, with an inhibition

constant ($K_i$) of 4.52 mM, whereas this substrate inhibition effect was not apparent in the WT (Fig. 5b, c and Supplementary Fig. 8).

ITC analysis failed to detect the binding of UMP to F161A, supporting the loss of inhibition by the nucleotide (Supplementary Table 2 and Supplementary Fig. 7). We also found that F161A binds one molecule of PALA ($K_D^{PALA} = 0.12$ μM) with 5-fold higher affinity than WT, in agreement with the enhanced inhibition, and also shows ~100-fold higher affinity for CP ($K_D^{CP} = 0.7$ μM) (Supplementary Table 2 and Supplementary Fig. 7). It seems likely that removal of the F161 side chain destabilizes the inhibited conformation of the CP-loop, reducing the affinity for UMP, and thus, favoring the alternate CP- or PALA-bound conformation (Fig. 4a, b).

To better understand the effect of the mutation, we determined the structure of F161A with UMP (Supplementary Table 1). In apparent contradiction with the activity and ITC results, the structure showed a molecule of UMP in the active site (Fig. 5d), the CP-loop in the inhibited conformation, and the missing F161 side chain being replaced by a glycerol molecule (Fig. 5e). It is probable that the low-affinity binding of the nucleotide to the active site of the mutated protein is favored by the higher concentrations of nucleotide (5 mM) and protein (135 μM) used in the crystallization condition compared to those in the ITC experiments (70 μM UMP at most and 30-40 μM protein). We also determined the structure of F161A crystallized with PALA (Supplementary Table 1). Interestingly, the structure showed a trimer bound to three molecules of PALA rather than one as in the WT (Fig. 5f and Supplementary Fig. 5). Although ITC indicated that PALA binds with high affinity to only one site per trimer (Supplementary Table 2), the high concentrations of PALA (2 mM) and protein in the crystallization condition must favor a low-affinity binding to the other subunits. Importantly, the three CP-loops fold in an active conformation, have well-defined electron density, and show no steric clashes around the molecular three-fold axis, since the bulky F161 side chain is missing and E160 adopts alternate conformations (Fig. 5g).

## Discussion

The pathway for de novo synthesis of UMP is evolutional conserved in all plants examined so far, and loss of function of any of the enzymes involved is presumably lethal. However, whereas downregulation of CPS, DHO, DHODH, or UMPS (Fig. 1a) had little or no effect[38,39], we showed that ATC downregulation strongly inhibits plant growth (Fig. 1b–e), as reported in previous studies[38,40], and causes a severe decrease in photosynthetic efficiency (Fig. 1f, g). Conversely, we also proved that plant growth can be enhanced by ATC overexpression (Fig. 1b–f). These results support the notion that ATC is not produced in large excess in the cell[38], and thus, those plants are especially sensitive to ATC levels that exert highest control over pyrimidine de novo synthesis. Indeed, the production of ATC is under transcriptional regulation in response to tissue pyrimidine availability[38,40] and to growth signals mediated by the TOR pathway[41]. However, transcription, synthesis and translocation of ATC into the chloroplast are slow and energetically costly processes that do not correct for rapid fluctuations needed to maintain nucleotide homeostasis. For this, allosteric regulation by UMP is the major mechanism controlling ATC activity in plants[7], but until now, we lack detailed information of how this feedback loop occurs.

Now, the structures of atATC reveal the mechanism of inhibition and explain the unsolved problem of why plant ATCs are inhibited by UMP and not by UTP as in other organisms[8,31] (Fig. 6). Rather than occupying an allosteric pocket, UMP binds and blocks the active site (Figs. 2 and 6a), directly competing with

CP, the substrate binding in first place[18,42]. UMP binds to the subunit in a wide-open conformation (Supplementary Fig. 3), where the N- and C-domains cannot move further apart to accommodate a di- or tri-phosphorylated nucleotide, thus explaining why UDP or UTP are not inhibitors. On the other hand, the pocket for the nitrogenous base is too small for the double ring of a purine and highly selective for uracil, since the methyl group of thymine would clash with the $^{349}PLP^{351}$ loop, whereas the cytidine amino group would distort the interaction with R310 (Fig. 2e). Also, one would expect the binding of deoxy-UMP to be weak based on the interaction between the ribose OH groups and the side chain of R248, which mimic the recognition of the Asp α-COOH group (Fig. 2e). However, these UMP-interacting elements are common to other ATCs that do not bind the nucleotide at the active site (Supplementary Fig. 1). Thus, we propose that the capacity of plant ATCs to be selectively inhibited by UMP relies on few small changes in the CP-loop (Figs. 2e and 4a). Indeed, a single point mutation in the CP-loop, F161A, is sufficient to turn atATC insensitive to UMP without affecting the catalytic efficiency of the enzyme nor its inhibition by PALA (Fig. 5, Supplementary Table 2, and Supplementary Fig. 8). Since the sequence of the CP-loop appears invariant in all plant ATCs known up to date (Supplementary Fig. 1), we propose that the UMP-inhibition mechanism described here for Arabidopsis ATC must be conserved across the plant kingdom. Thus, unlike other bacterial or eukaryotic ATCs that rely on complex associations with regulatory proteins (Fig. 6b, c), the current structures explain how plant ATCs have evolved to maintain a simple organization with both catalytic and regulatory capacities within a single protein chain (Fig. 6a). We hypothesize that this simplicity is a convenient solution for the transcriptional regulation and translocation of a single gene product into the chloroplast.

atATC has a surprising affinity for UMP, ~400-fold higher than for CP (Supplementary Table 2), which is explained by the extensive contacts of the nucleotide with the CP and Asp binding sites, similar to what PALA does. However, whereas binding of PALA involves large conformational changes (Fig. 3c), UMP binds to the more energetically favorable open state, and this might explain the 3-fold higher affinity for UMP than for the transition-state analog (Supplementary Table 2). It is uncertain whether such affinity for the natural inhibitor is retained in vivo or if it could be further increased by interaction with phospholipids, as reported for wheat-germ ATC[43]. In any case, based on these results, one would expect that ATC is constitutively repressed under steady-state conditions if UMP in the plastid reaches sub-millimolar concentrations similar to those estimated at the cytoplasm[38,39,44]. However, the concentration of UMP in the plastid is unknown and other features found in atATC suggest that this important activity is fine-tuned by the balance of UMP and CP contents in the cell. Indeed, we showed that the affinity for UMP is modulated by the communication between subunits, so that activity is diminished by high-affinity binding of the inhibitor to one subunit, whereas complete inactivation requires a 10-fold increase in the UMP pool (Supplementary Table 2). On the other hand, the response to UMP also varies with the concentration of CP, as shown by the change in the kinetic curves from a hyperbola to a sigmoid (Fig. 5a). This behavior, which was first described for wheat-germ ATC[7], could be relevant for the coordination of de novo pyrimidine and arginine synthesis, two pathways that depend on CP availability[11,12] (Fig. 1a). For instance, when pyrimidine pools are low, the activity of UMP-free ATC responds linearly with the concentration of CP (Fig. 5a, hyperbola), competing with arginine synthesis to ensure the production of nucleotides. Then, as UMP pool builds up, the synthesis of pyrimidines is reduced by the end product inhibition of ATC and CPS. However, upon CPS inhibition, ornithine

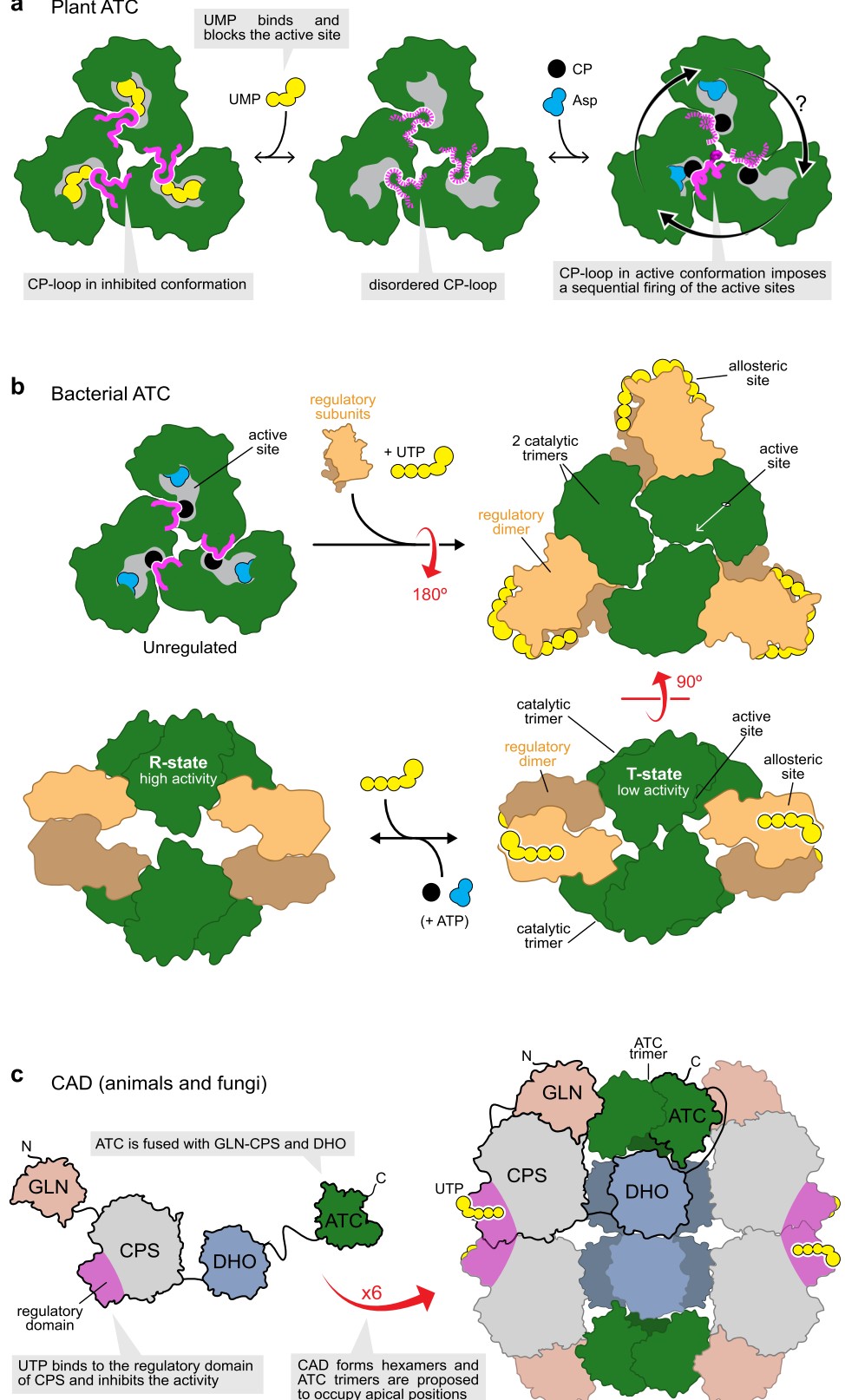

**Fig. 6 Different regulatory mechanisms in ATC. a** Plant ATCs present a unique mechanism of regulation, where UMP binds and blocks the active site. The CP-loop (represented in magenta) alternates between an UMP-bound inhibited conformation and an active conformation that ensures the sequential and perhaps ordered firing of the active sites. **b** In bacteria, isolated ATC catalytic trimers are unregulated. The association of two catalytic trimers with three dimers of regulatory subunits results in a holoenzyme that undergoes large conformational changes upon binding of UTP (inhibitor) or ATP (activator) to allosteric sites in the regulatory subunits. **c** In eukaryotes other than plants, ATC is fused together with CPS and DHO into a single multienzymatic protein named CAD that oligomerizes into hexamers where ATC trimers are proposed to occupy apical positions. The CPS and ATC activities are regulated by the binding of UTP to a regulatory region within the CPS domain.

accumulates and reverses the UMP inhibition on CPS, providing CP for arginine synthesis. Meanwhile, ATC activity remains negligible at low CP concentrations (Fig. 5a, sigmoid), but would sharply increase if the CP pool allows feeding both metabolic pathways.

atATC provides the example of an ATC trimer binding to a single molecule of PALA (Fig. 3b and Supplementary Table 2). Since PALA is a transition-state analog, these results strongly indicate that the reaction in plant ATCs might occur in only one subunit of the trimer at a time. Previously, we and others reported that human and *E. coli* ATCs exhibit much lower affinities for PALA by the third active site than the first two[24,45], leading us to suggest that despite being a trimer, ATCs might work more efficiently if not all the subunits were catalytically active at a time, perhaps avoiding intersubunit contacts that could slow down the conformational movements needed during catalysis[24] (Fig. 3c). Indeed, we proposed that the obstruction between subunits forced to work simultaneously could explain the inhibition at high substrate concentrations, a well-characterized but poorly understood phenomenon in ATCs[46]. Interestingly, we did not observe substrate inhibition in atATC (Figs. 4d, 5a and Supplementary Fig. 8), in agreement with the existence of a mechanism that ensures the reaction of one subunit at a time. This mechanism relies on the projection of F161 towards the three-fold axis, which prevents the CP-loops from reaching simultaneously the active conformation (Fig. 4b, d). Thus, F161 plays a dual role both stabilizing the UMP inhibited conformation and synchronizing the firing of the subunits. Indeed, mutation F161A does not only turn the enzyme insensitive to UMP but also allows the binding of three PALA molecules per trimer (Fig. 5e and Supplementary Table 2), suggesting that the reaction can occur simultaneously in the three active sites. However, the activity of F161A is not 3-fold higher than the WT, likely due to the obstruction between subunits acting simultaneously. Indeed, we observed substrate inhibition for F161A (Fig. 5b, c and Supplementary Fig. 8), although less acute than other non-plant ATCs[42,46].

Lastly, it is tempting to speculate that the sequential activation of the subunits in plant ATCs could follow a specific order (Fig. 6a). Certainly, the atATC structure with PALA and CP provides a suggestive snapshot of each subunit at a different stage of the reaction (Fig. 4c), and further studies should explore this possibility.

The de novo pyrimidine synthesis pathway is a promising target for biomedical and biotechnological intervention[47,48]. Our analysis uncovers unique regulatory and catalytic mechanisms of plant ATCs, offering the possibility to modulate this central enzymatic activity and selectively inhibit or enhance plant growth in a similar manner as observed by changing protein expression (Fig. 1b). For instance, the design of new herbicides could be based on UMP analogs with additional groups replacing the observed water-mediated interactions (Fig. 2e), or on compounds targeting the cavity at the three-fold axis and blocking the sequential activation of the CP-loops. On the other hand, it should be possible to enhance growth of transgenic crops expressing UMP-insensitive and thus, constitutively active ATC. The present results should guide in the design of highly efficient and non-regulated ATC variants of biotechnological interest.

## Methods

**Plant growth**. For DNA isolation, tissue collection and phenotypic inspection, wild-type *Col*-0 and transgenic *Arabidopsis thaliana* (L.) Heynh. plants (ecotype Columbia) were used throughout. Plants were grown in standardized soil (ED73 Einheitserde, Humuswerke Patzer) soil under long day conditions (120 μmol quanta m$^{-2}$ s$^{-1}$ in a 12 h light and 12 h dark regime, temperature 22 °C, humidity 60%). Prior to germination, seeds were incubated for 24 h in the dark at 4 °C for imbibition. Alternatively, to assess the effect of PALA on growth, plants were cultivated under short-day conditions in liquid culture according to a protocol

suitable for fresh weight determination and feeding of effector molecules[49]. For growth experiments under sterile conditions, the seeds were surface sterilized in 5% sodium hypochloride before adding them to the ½ MS liquid medium supplemented with 1% (w/v) sucrose with or without 0.1 mM, 0.2 mM, or 0.4 mM PALA. The liquid cultures were maintained on a shaker at 100 rpm under the same light and temperature conditions as soil-grown plants. Leaf extract of wild-type and mutants was prepared by homogenizing leaf material in extraction buffer (50 mM HEPES-KOH pH 7.2, 5 mM MgCl$_2$, 2 mM phenylmethylsulfonyl fluoride (PMSF) on ice. The homogenous extract was centrifuged at 20,000$g$ for 10 min at 4 °C. The supernatant was collected and stored on ice until use.

**Construction of ATC knock down and overexpressor plants**. ATC (*pyrB*; At3g20330) knock down mutants were generated using an established protocol for gene silencing by artificial microRNA (amiRNA)[50]. An amiRNA targeting *pyrB* was designed using an online tool (http://wmd3.weigelworld.org). The sequence *TAATGACAGGTATATCGGCAG* was used for generation of primers, and Gateway™ compatible sequences attP1 and attP2 were added to primers to engineer the amiRNA fragment (Supplementary Table 3). Subsequently, the fragment was subcloned via BP-clonase reaction into the Gateway™ entry vector pDONR/Zeo and via LR-clonase reaction into the destination vector pK2GW7, which contains a 35S-CaMV promoter. Several independent lines were obtained exhibiting 16-10% of transcript and two were selected for further analysis. *ATC* overexpressor plants were generated by cloning full-length *ATC* (for primers see Supplementary Table 3) using Gateway technology into pUB-Dest under the control of the ubiquitin-10 promoter[51].

**Gene expression analyses**. Leaf material of soil-grown plants was collected and homogenized in liquid nitrogen prior to extraction of RNA with the Nucleospin RNA Plant Kit (Macherey-Nagel, Düren, Germany) according to the manufacturer's advice. RNA purity and concentration were quantified using a NanoDrop spectrophotometer. Total RNA was transcribed into cDNA using the qScript cDNA Synthesis Kit (Quantabio, USA). qPCR was performed using the quantabio SYBR green quantification kit (Quantabio) on PFX96 system (BioRad, Hercules, CA, USA) using specific primers (Supplementary Table 3), and At2g3760 (Actin) was used as reference gene for transcript normalization.

**Pulse-amplitude-modulation (PAM) fluorometry measurements**. A MINI-IMAGING-PAM fluorometer (Walz Instruments, Effeltrich, Germany) was used for in vivo chlorophyll A light curve assays on intact, 6 week old dark-adapted plants using standard settings[52].

**Protein production**. Arabidopsis ATC sequence encoding aa 82–390 was PCR amplified from cDNA using a pair of specific primers (Supplementary Table 3) and transferred via NdeI-XhoI to a modified pET28a plasmid[53]. Mutant F161A was made using the Quick-Change II-E site-directed mutagenesis kit (Stratagene) and a pair of complementary primers (Supplementary Table 3). Transformed *E. coli* BLR (DE3)-pLysS cells (Merck) were grown overnight at 37 °C in Terrific Broth (TB) medium containing 25 μg ml$^{-1}$ kanamycin, 10 μg ml$^{-1}$ tetracycline, and 25 μg ml$^{-1}$ chloramphenicol to an optical density at 600 nm of 0.7–0.9. Protein expression was induced with 0.5 mM isopropyl β-D-thiogalactopyranoside (IPTG) overnight at 37 °C. Cells were resuspended in buffer A (20 mM Tris–HCl pH 8.0, 0.5 M NaCl, 10 mM imidazole, 5% glycerol, 2 mM β-mercaptoethanol) and disrupted by sonication. The clarified lysate was incubated with Ni-sepharose 6 fast flow beads (GE Healthcare) equilibrated in buffer A. After washing with buffer A supplemented with 40 mM imidazole, the protein was eluted in buffer A with 300 mM imidazole. The protein was further purified by size-exclusion chromatography on a Superdex 75 10/300 column (GE Healthcare, USA) equilibrated in buffer GF [20 mM Tris pH 7.0, 0.1 M NaCl, 2% glycerol, 0.2 mM Tris(2-carboxyethyl) phosphine (TCEP)]. The protein was concentrated to 5 mg ml$^{-1}$ in an Amicon ultracentrifugation device (Millipore) with 10 kDa cutoff and directly used for further studies or supplemented with 40% glycerol, flash-frozen in liquid nitrogen and stored at −80 °C. All purification steps were carried out at 4 °C and the purity of the sample was evaluated by SDS-PAGE and Coomassie staining. Protein concentration was determined by Bradford assay[54] using bovine serum albumin (Sigma) for the standard curve.

To remove UMP from the purified protein, the sample was diluted to 1 mg ml$^{-1}$ and supplemented with 50 mM CP and 100 mM Asp, and filtered at room temperature through three consecutive PD-10 desalting columns (GE Healthcare) equilibrated in GF buffer containing 50 mM CP and 100 mM Asp. In between columns, the 3.5 ml eluted sample was concentrated down to 2.5 ml using an Amicon ultracentrifugation device. In the last step, the sample was filtered through a PD-10 in GF buffer without substrates.

**SEC-MALS analysis**. Molar mass was determined by size-exclusion chromatography coupled to multi-angle light scattering (SEC-MALS). In all, 400 μl of purified protein at 1.4 mg ml$^{-1}$ was fractionated on a Superdex 200 10/300 column equilibrated in GF buffer using an AKTA purifier (GE Healthcare) at a flow rate of 0.5 ml min$^{-1}$. The eluted sample was characterized by in-line measurement of the refractive index and multi-angle light scattering using Optilab T-rEX and

DAWN 8+ instruments, respectively (Wyatt). Data were analyzed with ASTRA 6 software (Wyatt) to obtain the molar mass and plotted with software GraphPad.

**Immunoblotting**. In all, 15 ng of recombinant ATC or 15 μg of a fresh protein extract from Arabidopsis leaves separated in a 15% SDS-PAGE gel were transferred onto a nitrocellulose membrane (Whatman, Germany) by wet blotting. The membrane was blocked in phosphate-buffered saline plus 0.1% (v/v) Tween 20 (PBS-T) with 3% milk powder for 1 h at room temperature, followed by three washes of 10 min in PBS-T. Then, the membrane was incubated with a rabbit polyclonal antiserum raised against recombinant ATC (Eurogentec, Belgium) for 1 h, followed by three washes with PBS-T. Next, the membrane was incubated for 1 h with a horseradish peroxidase (HPR) conjugated anti-rabbit antibody (Promega, Walldorf, Germany) diluted in PBS-T with 3% milk powder. The result was visualized by chemiluminescence using the ECL Prime Western blotting reagent (GE Healthcare) and a Fusion Solo S6 (Vilber-Lourmat) imager.

**Crystallization and structure determination**. Initial crystallization screenings were performed at room temperature using the sitting-drop vapor diffusion method and 96-well MRC plates (Hampton). Drops consisting of 0.7 μl protein solution at 5 mg ml⁻¹ plus 0.7 μl reservoir solution were equilibrated against 60 μl of reservoir solution using JCSG+, PACT, and Crystal Screen (Hampton Research) commercial screens. Diamond and rod-shape crystals appeared after few hours. Initial hits were further optimized in 24-well sitting-drop plates (Qiagen) using as reservoir solution 18–22 % PEG 3350, 0.1 M Na₂SO₄ and 0.1 M Bis-Tris pH 6.5. The protein freed of UMP was crystallized in 18–22% PEG 3350 and 150–200 mM potassium acetate. Also, the protein was co-crystallized with 2 mM PALA or 10 mM CP in 25% PEG 3350, 0.2 M Li₂SO₄, and 0.1 M Bis-Tris pH 5.5. Crystals of mutant F161A were obtained in similar conditions as the WT protein. All crystals were cryo-protected in mother liquor supplemented with 20% glycerol and flash-frozen in liquid nitrogen. X-Ray diffraction data of the APO and +PALA crystals were collected at beamline MASSIF (ESRF, Grenoble) and the other crystals were diffracted at beamline BL13-XALOC (ALBA synchrotron, Barcelona) at specific wavelengths (APO and +PALA, 0.966 Å; +UMP, 0.9793 Å; +PALA + CP, 0.97924 Å; +CP, 0.97910 Å, F161A + UMP and F161A + PALA, 0.97926 Å) and at 100 K temperature using Pilatus 6 M detectors (DECTRIS). Data processing and scaling were performed with XDS[55] and autoPROC[56]. Crystallographic phases were obtained by molecular replacement using PHASER[57] and the structure of the ATC domain of human CAD (PDB 5G1O)[24] as the search model. The models were constructed by iterative cycles of model building in COOT[58] and refinement with PHENIX[59] or Refmac5[60] in CCP4[61]. Data collection and refinement statistics are summarized in Supplementary Table 2. In all structures, over 97% residues are in the favored region of the Ramachandran plot and only the APO and +CP structures contain, respectively, 0.11% and 0.05% Ramachandran outliers adjacent to disorder regions. Figures were prepared with PyMOL.

**Isothermal titration calorimetry**. Experiments were performed in an Auto-iTC200 calorimeter (MicroCal, Malvern-Panalytical) at 25 °C with a 0.2 ml reaction cell and 30–40 μM protein solution (in protomer units) in the cell. Titration experiments consisted of 19 injections of 2 μl of 0.4 mM PALA, 0.4 mM UMP or 1 mM CP. Binding of UMP or CP was also performed on 30–40 μM protein samples pre-incubated with 80 μM PALA. All solutions were in GF buffer and degassed and mixed in the cell by stirring at 750 rpm. Data analysis was performed with Origin 7 (OriginLab) using a general three-site cooperative and a non-cooperative binding model, restricted to one or two ligand binding sites when convenient[24,62,63]. Non-linear least-squares regression analysis was employed to estimate dissociation constants for the interaction of the different ligands with ATC. The number of ligand binding sites or stoichiometry was fixed by the model employed for the analysis (one site, two sites, or three sites); in all experiments the fraction of active or binding-competent protein was close to 1.

**Activity assays**. Activity was assayed by a colorimetric method[64] adapted to a 96-well plate format[24]. The reaction was carried out in 50 mM Tris-acetate (pH 8.3) and 0.1 mg ml⁻¹ BSA in a final volume of 150 μl. atATC was pre-incubated with Asp for 10 min in a water bath at 25 °C and the reaction was triggered by adding CP and stopped after 5 min with 100 μl of a color solution consisting of two parts of reagent A [0.37% (w/v) antipyrine and 0.25% (w/v) ammonium iron(III) sulfate in 25% (v/v) H₂SO₄ 95% and 25% (v/v) H₃PO₄ 85%] and one part of reagent B [0.4% (w/v) diacetylmonoxime in 7.5% (w/v) NaCl]. Reagent B is light sensitive and was stored at 4 °C in the dark. The color reagents were gently mixed and stored at 4 °C in the dark until added to the sample. Reaction tubes were closed, boiled at 95 °C for 15 min, and kept in the dark for 30 min before measuring the absorbance at 450 nm in a Tecan infinite m200 microplate reader (Tecan). Substrate saturation curves were done varying the concentration of one substrate maintaining a fix concentration of the other substrate (5 mM for CP or 10 mM for Asp). Data analysis was done with GraphPad. In the absence of inhibitors, WT kinetics obey Michaelis-Menten equation $v = [V_{max}·X/K_{0.5} + X]$, where $X$ is substrate concentration and $K_{0.5}$ is the substrate concentration at which the initial rate ($v$) is one-half of the maximum velocity ($V$max)]. In the absence of UMP or PALA, F161A shows inhibition by excess of substrate and the data was fitted to equation

$v = V_{max}·X/K_{0.5} + X(1 + X/K_i)$, where $K_i$ is the inhibition constant. In the presence of UMP or PALA, WT kinetics obey the Hill equation [$v = V_{max}·X^h/K_{0.5}{}^h + X^h$], whereas F161A kinetics were fitted to a modified equation with the additional term for substrate inhibition: $v = V_{max}·X^h/K_{0.5}{}^h + [X^h(1 + X/K_i)]$. Best-fitting values for the latter equation are ambiguous, since many combinations of these parameters generate curves that fit the data equally well. Protein concentration in the assay was 0.3 μM (0.01 mg ml⁻¹) both for WT and F161A. Inhibition was measured by adding PALA or UMP during the pre-incubation.

**Reporting summary**. Further information on research design is available in the Nature Research Reporting Summary linked to this article.

## Data availability

The structural data (coordinates and structure factors) that support the findings of this study are deposited in the Protein Data Bank (PDB) under accession codes 6YPO (+UMP), 6YY1 (APO), 6YS6 (+PALA), 6YSP (+PALA + CP), 6YVB (+ CP), 6YWJ (F161A + UMP), and 6YW9 (F161A + PALA). The ITC data that support the findings of this study are available from the corresponding author upon reasonable request. A separate source data file contains raw data underlying Figs. 1c–g, 2b, c, 3a, 5a–c, and Supplementary Figs. 4b, c and 8; it also contains full blots and gels of cropped images shown in Figs. 1 and 2 (see Supplementary Fig. 9). Source data are provided with this paper.

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

## Acknowledgements

This work was supported by funding from the Spanish Ministry of Science, Innovation and Universities (BFU2016-80570-R and RTI2018-098084-B-100; AEI/FEDER, UE) to S.R.-M., DFG (Mo 1032/4-1 and CRC175-B08) to T.M., DFG-IRTG 1830 to T.M. and L.B., and Bayer AG Crop Science (Grants for Targets 2017-02-005) to S.R.-M. and T.M. We thank the staff from ALBA (Barcelona, Spain) and ESRF (Grenoble, France) synchrotron facilities for help during crystallographic data collection; R. Campos-Olivas and C.M. Santiveri for support with SEC-MALS analyses; S. Niopek-Witz and N. Navaseelan for support with ATC expression and kinetic analysis; L. Ohler, A. John, and A. Grande-García for support during protein purification, mutant generation, and crystallization trials; and M. Moreno-Morcillo for support with crystal structure analysis. We thank W. Yang, D. Lietha, and E. Neuhaus for critical reading of the manuscript.

## Author contributions

T.M and S.R.-M. conceived the project and all authors contributed to design of experiments. L.B. and T.M. generated and analyzed ATC mutant lines. L.B., F.D.C.-O., and S.R.-M. purified and crystallized the proteins and determined the crystal structures. A.V.-C. performed ITC experiments. L.B. carried out the enzymatic assays and S.R.-M.

analyzed the data. S.R.-M. and T.M. wrote the manuscript. All authors were involved in the discussion for data analysis and commented on the manuscript.

## Funding

## Competing interests

The authors declare no competing interests.
