## [Peer Review File · Nature Communications]

REVIEWER COMMENTS

Reviewer #1 (Remarks to the Author):

The authors achieved to obtain structural information of the Arabidopsis aspartate transcarbamoylase (atATC) without bound ligand, with bound transition state mimic PALA and with bound inhibitor UMP. Overall the structure is similar to ATCs from other organisms (E. coli, human) with some noteworthy differences. Trimeric AtATC is inhibited by high affinity binding of UMP to the carbamoyl phosphate binding site (other ATCs are inhibited by UTP) and binds only one PALA (not up to three as in other ATCs) to that same site. It has been noted before that plant ATC is inhibited by UMP but with the structure the molecular basis for this inhibition can now be explained. The authors suggest that the binding behavior of PALA indicates that the ATC subunits are active in a sequential manner and not in parallel.

The authors identify a critical Phe residue (F161) that is required for high affinity UMP binding and for preventing more than one PALA to bind. A F161A variant enzyme is not inhibited by UMP anymore and binds up to three PALA molecules.

The authors also show that downregulation of atATC by RNAi leads to dwarfed plants with decreased photosynthetic activity, which can be mimicked by PALA supplements to the wild type, whereas over-expression of atATC increases plant size. The effects of plant ATC downregulation have been described before. Chen and Slocum (2008) produced RNAi-plants of atATC and showed their decreased biomass. They also demonstrated that the phenotype can be mimicked with PALA. Basset et al. (2003) demonstrated that PALA-treated plants are chlorotic and have less chloroplasts. However, to my knowledge it has not yet been shown that ATC over expression can increase plant size.

The inhibition of plant ATCs by UMP is long known (e.g. Neuman and Jones, 1962, Doremus and Jagendorf, 1995) and a detailed kinetic characterization of this inhibition has been undertaken (Khan et al., 1999). However, the authors can now relate these findings to the structural basis for UMP inhibition in plant ATCs, which is the main novelty of this report. The relief of the inhibition in the F161A mutant underscores the validity of the structural findings.

It would have been interesting to see the effects of an F161A over expression line on plant growth. Can such a transgene also increase plant size, even when only moderately expressed and not strongly over produced? The F161A mutant expressed by the ATC promoter could be introduced into a heterozygous ko line and homozygous ko lines carrying the transgene could be selected. I feel that such a validation of the structural findings in vivo would strongly increase the significance of this work, and it would be very interesting to see how deregulated ATC influences plant physiology. Obviously, it would also be interesting to learn how the amounts of (pyrimidine) nucleotides are disturbed in vivo in the different presented lines (and an F161A line). Is the increased growth in the overexpression line simply explained by higher concentrations of pyrimidine nucleotides?

In the discussion I was missing some speculation regarding the particular differences of plant ATC regulation versus E. coli or human ATC regulation. Why do plants use UMP instead of UTP for regulation?

Minor points:

Correct reference 13. Something is wrong there.

P6, 17 should read seedlings (not 'seeds')

Reviewer #2 (Remarks to the Author):

In this paper, Bellin and colleagues report the first x-ray crystal structure of an aspartate transcarbamoylase, an enzyme at the interface of amino acid metabolism and nucleotide metabolism, from a plant and identified a single residue responsible for modulating regulation by physiological and synthetic inhibitors. Unlike mammalian and microbial ATCs, the plant enzyme does not have associated regulatory proteins and has evolved the ability to regulate its own activity without the need for additional proteins. The authors thoroughly investigated the evolution of function and regulation in this unique enzyme using plant genetics, x-ray crystallography, site-directed mutagenesis, ITC, and kinetics. The data they present are exciting, high quality, of broad interest to metabolic biochemists, and greatly contributes to our understanding of pyrimidine metabolism in plants. My comments are as follows:

1. Have you or others determined the IC50 for UMP and PALA? It would be interesting to know how this value relates to the physiological concentrations of UMP in the plastid. You mention on pg. 11, line 9 that the UMP reaches sub-millimolar concentrations, but is this only true for cytosolic UMP?
2. Could you please include the Gene ID for ATC?
3. For your seedling assays, please include information about the number of seedlings you measured.
4. On pg. 16, line 8, could you include information about how you generated your standard curve for your Bradford assay? Using BSA?
5. On pg. 18, line 13, please provide more information about the "color solution" used for this assay.

-Cynthia Holland

Reviewer #3 (Remarks to the Author):

In the manuscript "Mechanisms of feedback inhibition and sequential firing of active sites in plant aspartate transcarbamoylase," Bellin et al. thoroughly investigated structures of plant aspartate transcarbamoylase (ATC) in the de novo pyrimidine biosynthetic pathway using a combination of diverse experimental techniques, including in vivo plant analysis and biophysical analysis. Since the de novo pyrimidine synthesis pathway has potential value for a variety of pharmacological and agricultural applications, understanding the regulatory and catalytic mechanisms of plant ATCs is able to contribute to the research field of plant biotechnology significantly. Although the authors have proposed unique regulatory mechanisms based on multiple crystal structures and biophysical experiments, they must provide more details to support their hypotheses.

- ATC is key for plant development and photosynthetic efficiency

Page 4., Lines 17-26 & Figure 1. 3)

 The authors measured ATC transcript levels in knockdown and overexpressing lines (n = 3). More numbers of plants need to be tested.

- Crystal structure of Arabidopsis ATC bound to UMP

Table 1.

 While most of the crystal structures are well processed, some of the statistics question data reliability. In particular, the I/sigma value of 6YSP (+ PALA + CP), 10.71 (1.35) , is very low. Please reprocess the data set or provide other parameters of the highest resolution shell.

Page 5. Line 26

In addition, atATC has the CP-loop (aa 156–169) and Asp-loop (aa 309–332) (Fig. 2a,d) that in other ATCs undergo large conformational movements upon substrate binding.

 The authors need to display detailed information about the large conformational movement in the figure.

Page 5., Line 29, Fig. 2d,e and Supplementary Fig. 1

 The electron density maps are provided in Supplementary Fig 2.

atATC only binds one molecule of PALA per trimer

Page 6. Lines 14-19.

 The authors showed the effect of PALA [N-(phosphonacetyl)-L-aspartate] on root development. However, the number of plant samples and the error values are not indicated.

Page 6., Line 23

Surprisingly, the structure showed the atATC trimer with PALA bound to only one of the subunits (Fig. 3b).

 As shown in Supplementary Fig 2., provide the electron density maps of 1. PALA, 2. empty, and 3. two sulfate ions and one glycerol molecule in each monomer. Provide the B-factor value of PALA.

Page 7., Line 10,

Isothermal titration calorimetry (ITC) analyses

 The authors should provide all parameters of their ITC experiment results, including ΔG (kcal/mol), ΔH (kcal/mol), $-\Delta S$ (kcal/mol), and error values. There is currently insufficient data to support the author's hypothesis because K_d values alone cannot determine the quality of the data.

Figure 5. e & g Figure 4 a. & b

 Use consistently defined secondary structures (e.g., helices) to compare the same regions of two structures.

Mutation F161A abolishes UMP inhibition and allows binding of three PALA molecules

Page 8., Line 16

 Detailed kinetic analysis is required to confirm the effect of mutation F161A on the enzymatic activity. Provide steady-state kinetic parameters in a table, including enzyme efficiency, to compare WT and mutant activities.

Page 9., Lines 9-11

Although ITC indicated that PALA binds with high affinity to only one site per trimer (Table 2), the high concentration of PALA (2 mM) in the crystallization condition must favor a low-affinity binding to the other subunits.

 Given the low binding affinity in Table2, it is difficult to explain the three ligand bindings in the crystal structure. In other words, if the three PALA bindings can be seen at a concentration of 2mM, the authors could be able to calculate the Kd value in the ITC experiment.

Reviewer #1

The authors achieved to obtain structural information of the Arabidopsis aspartate transcarbamoylase (atATC) without bound ligand, with bound transition state mimic PALA and with bound inhibitor UMP. Overall the structure is similar to ATCs from other organisms (E. coli, human) with some noteworthy differences. Trimeric AtATC is inhibited by high affinity binding of UMP to the carbamoyl phosphate binding site (other ATCs are inhibited by UTP) and binds only one PALA (not up to three as in other ATCs) to that same site. It has been noted before that plant ATC is inhibited by UMP but with the structure the molecular basis for this inhibition can now be explained. The authors suggest that the binding behavior of PALA indicates that the ATC subunits are active in a sequential manner and not in parallel. The authors identify a critical Phe residue (F161) that is required for high affinity UMP binding and for preventing more than one PALA to bind. A F161A variant enzyme is not inhibited by UMP anymore and binds up to three PALA molecules. The authors also show that downregulation of atATC by RNAi leads to dwarfed plants with decreased photosynthetic activity, which can be mimicked by PALA supplements to the wild type, whereas over-expression of atATC increases plant size. The effects of plant ATC downregulation have been described before. Chen and Slocum (2008) produced RNAi-plants of atATC and showed their decreased biomass. They also demonstrated that the phenotype can be mimicked with PALA. Basset et al. (2003) demonstrated that PALA-treated plants are chlorotic and have less chloroplasts. However, to my knowledge it has not yet been shown that ATC over expression can increase plant size. The inhibition of plant ATCs by UMP is long known (e.g. Neuman and Jones, 1962, Doremus and Jagendorf, 1995) and a detailed kinetic characterization of this inhibition has been undertaken (Khan et al., 1999). However, the authors can now relate these findings to the structural basis for UMP inhibition in plant ATCs, which is the main novelty of this report. The relief of the inhibition in the F161A mutant underscores the validity of the structural findings.

We thank Reviewer #1 for the thorough revision of the manuscript, and for noticing the novelty of the study.

1. It would have been interesting to see the effects of an F161A over expression line on plant growth. Can such a transgene also increase plant size, even when only moderately expressed and not strongly over produced? The F161A mutant expressed by the ATC promoter could be introduced into a heterozygous ko line and homozygous ko lines carrying the transgene could be selected. I feel that such a validation of the structural findings *in vivo* would strongly increase the significance of this work, and it would be very interesting to see how deregulated ATC influences plant physiology.

We agree with Reviewer #1 on the great interest of studying the over expression of F161A mutant on plant growth, and we appreciate the indications on how the transgene plant could be obtained. Using a similar approach, we already initiated the construction of such plant mutant. However, we expect that the identification of a heterozygous ATC KO mutant, re-transformation, backcrossing and analysis of the plant lines will still require at least 8-12-month work. Thus, it is our intention to report these *in vivo* results in a follow-up work.

2. Obviously, it would also be interesting to learn how the amounts of (pyrimidine) nucleotides are disturbed *in vivo* in the different presented lines (and an F161A line). Is the increased growth in the overexpression line simply explained by higher concentrations of pyrimidine nucleotides?

We agree with Reviewer #1 on the interest of measuring the amounts of pyrimidines. The quantification of nucleotides in plants has been long time hampered by the complexity of secondary metabolites in plant extracts and by the difficulties of extraction and analysis. Only recently, new methods have been optimized to perform these studies. In line with the Reviewer's suggestion, we performed an extensive study to quantify a large set of metabolites including pyrimidines in ATC knock-down lines. In parallel, we performed a similar quantification on knock-down lines for DHODH, another enzyme involved in pyrimidine biosynthesis. This extensive study, performed in close collaboration with a team of experts in analytical techniques, has shown that UMP, UDP, different intermediates in the synthesis of pyrimidines and UDP-glucose were significantly depleted in ATC knock-down lines (see figure below). In consequence, the plants present reduced photosynthesis, accumulation of reactive oxygen

species and a low energy status. This thorough metabolite analysis of ATC and DHODH downregulated plants is the subject of a manuscript that is ready for submission. Therefore, we would have problems to transfer part of this data to the manuscript under revision, which is

intended to provide a structural insight to the physiology of ATC in a limited way, focusing on the mechanistic aspects.

Metabolites levels from fully developed leaves in ATC and DHODH downregulated lines. Selected metabolites are shown, Col-0 (black bars) was set to 1. Metabolite levels are given as log₂-fold changes relative to Col-0. Asterisks indicate significantly (p -value < 0.05, Wilcoxon Mann-Whitney) altered levels. $n=7$.

We also agree with Reviewer #1 that a similar metabolome analysis for the ATC overexpressor lines is of great interest. Indeed, the work is already in progress, and we hope that it will be the subject of a follow-up article. We hypothesize that increased pyrimidine levels will most likely provoke increased growth. A crosstalk between nucleotide levels and the activity of TOR, as central growth regulator, is becoming more and more visible not only in animals but also in plants.

3. In the discussion I was missing some speculation regarding the particular differences of plant ATC regulation versus *E. coli* or human ATC regulation. Why do plants use UMP instead of UTP for regulation?

Reviewer #1 points to a very interesting question. The current structural data provides solid evidence that UMP inhibits because it fits in the active site, whereas UTP doesn't. This is explained in the discussion, and we have modified it to make the explanation clearer (see below). However, beyond this undeniable answer, we are still intrigued by some other aspects that could also influence the usage of UMP rather than UTP, such as the transport of nucleotides into the plastid, or the possibility that UMP could be synthesized by the pyrimidine salvage pathway within the chloroplast. These questions go beyond the scope of this article, and we hope to pursue in future work.

(page 10, line 11 - page 11, line 4) Now, the structures of atATC reveal the mechanism of inhibition and explain the unsolved problem of why plant ATCs are inhibited by UMP and not by UTP as in other organisms^{8,31} (Fig. 6). Rather than occupying an allosteric pocket, UMP binds and blocks the active site (Figs. 2 and 6a), directly competing with CP, the substrate binding in first place^{18,42}. UMP binds to the subunit in a wide-open conformation (Supplementary Fig. 3), where the N- and C-domains cannot move further apart to accommodate a di- or tri-phosphorylated nucleotide, thus explaining why UDP or UTP are not inhibitors. On the other hand, the pocket for the nitrogenous base is too small for the double ring of a purine and highly selective for uracil, since the methyl group of thymine would clash with the ³⁴⁹PLP³⁵¹ loop, whereas the cytidine amino group would distort the interaction with R310 (Fig. 2e). Also, one would expect the binding of deoxy-UMP to be weak based on the interaction between the ribose OH groups and the side chain of R248,

which mimic the recognition of the Asp α -COOH group (Fig. 2e). However, these UMP-interacting elements are common to other ATCs that do not bind the nucleotide at the active site (Supplementary Fig. 1). Thus, we propose that the capacity of plant ATCs to be selectively inhibited by UMP relies on few small changes in the CP-loop (Figs. 2e and 4a). Indeed, a single point mutation in the CP-loop, F161A, is sufficient to turn atATC insensitive to UMP without affecting the activity nor the inhibition by PALA (Fig. 5 and Table 2). Since the sequence of the CP-loop appears invariant in all plant ATCs known up to date (Supplementary Fig. 1), we propose that the UMP-inhibition mechanism described here for *Arabidopsis* ATC must be conserved across the plant kingdom. Thus, unlike other bacterial or eukaryotic ATCs that rely on complex associations with regulatory proteins (Fig. 6b,c), the current structures explain how plant ATCs have evolved to maintain a simple organization with both catalytic and regulatory capacities within a single protein chain (Fig. 6a). We hypothesize that this simplicity is a convenient solution for the transcriptional regulation and translocation of a single gene product into the chloroplast.

4. Correct reference 13. Something is wrong there.

We thank Reviewer #1 for noticing the error in this reference. We corrected it as follows:

(page 20, line 30-31) 13. Lovatt, C. J. & Cheng, A. H. Aspartate carbamyltransferase: site of end-product inhibition of the orotate pathway in intact cells of *Cucurbita pepo*. *Plant Physiol* **75**, 511–515 (1984).

5. P6, 17 should read seedlings (not 'seeds')

Thank you for the correction. It has been modified as suggested (in page 6, line 19).

Reviewer #2

In this paper, Bellin and colleagues report the first x-ray crystal structure of an aspartate transcarbamoylase, an enzyme at the interface of amino acid metabolism and nucleotide metabolism, from a plant and identified a single residue responsible for modulating regulation by physiological and synthetic inhibitors. Unlike mammalian and microbial ATCs, the plant enzyme does not have associated regulatory proteins and has evolved the ability to regulate its own activity without the need for additional proteins. The authors thoroughly investigated the evolution of function and regulation in this unique enzyme using plant genetics, x-ray crystallography, site-directed mutagenesis, ITC, and kinetics. The data they present are exciting, high quality, of broad interest to metabolic biochemists, and greatly contributes to our understanding of pyrimidine metabolism in plants.

We thank Reviewer #2 for the positive comments and careful revision of the manuscript.

My comments are as follows:

1. Have you or others determined the IC50 for UMP and PALA? It would be interesting to know how this value relates to the physiological concentrations of UMP in the plastid. You mention on pg. 11, line 9 that the UMP reaches sub-millimolar concentrations, but is this only true for cytosolic UMP?

We thank the reviewer for the suggestion to compare the IC50 of UMP and PALA with the physiological concentrations of UMP in the plastid. This is indeed a key question, which refers to how the enzyme behaves in the natural context. Earlier studies, measured metabolites in isolated chloroplasts. However, we know now that many metabolites leak out during the isolation process and cannot be quantified correctly by this method. Currently, the quantification of ribonucleotides in the chloroplast is possible using a combination of specific sensors (only available for ATP) and NAF (non-aqueous fractionation) coupled to LC-MS, but this approach has not been used to study pyrimidine levels in plants. To the best of our knowledge, nowadays, nobody knows what the actual concentration of nucleotides in the plastids is: “Not much is known about (1) the phosphorylation state in which nucleotides are taken up, (2) which transporters are involved, or (3) whether the concentrations of (deoxy) nucleotides differ in the distinct cellular compartments and how this may be regulated—subcellular distributions have

been estimated only for the adenylates” (Witte, C. P. & Herde, M. *Nucleotide Metabolism in Plants*. 2020 *Plant Physiol* 182, 63–78). Thus, the only data available are those studies cited in the manuscript that refer to cytosolic nucleotide levels. We have modified the discussion to make this point clearer:

(page 11, lines 11-16) In any case, based on these results, one would expect that ATC is constitutively repressed under steady state conditions if UMP in the plastid reaches sub-millimolar concentrations similar to those estimated at the cytoplasm^{38,39,44}. However, the concentration of UMP in the plastid is unknown and other features found in atATC suggest that this important activity is fine-tuned by the balance of UMP and CP contents in the cell.

Regarding the IC50 values, other studies have reported IC50 values that are not very consistent. This is likely due to the fact that IC50 is an apparent measurement that depends on the conditions of the enzymatic assay, including the concentration of enzyme and the mechanism of inhibition. Rather than the IC50, we have measured by ITC the dissociation constant (K_D), which provides a direct measurement of the strength of interaction of UMP with the enzyme.

2. Could you please include the Gene ID for ATC?

Thank you for the suggestion. Although this information was included in the Methods (page 14, line 21), we now provide it in a more visible place, within the Results section at first mentioning:

(page 4, line 19) To explore the importance of ATC for plant growth we used artificial microRNA (amiRNA) to knockdown ATC (At3g20330) in Arabidopsis.

3. For your seedling assays, please include information about the number of seedlings you measured.

Thank you for the suggestion. The number of seedlings was n=3 for fresh-weight (we collected 10 seedlings for one measurement because of the small size, and did this in triplicate) and n=30 for root length. We included further experimental results in root length determination and calculated corresponding errors. We have included this information in the text as follows:

(page 6, lines 18-22) Seedling assays in presence of 0.2 mM or 0.4 mM PALA showed a decrease in fresh weight to 59% or 23%, respectively, compared to untreated seedlings (3 times 10 seedlings were weighted per treatment, n=3). Root length in untreated seedlings was 2.68 cm \pm 0.49 and was reduced to 36% at 0.2 mM PALA (0.96 cm \pm 0.22) and 8% at 0.4 mM PALA (0.21 cm \pm 0.09)(n=30) (Fig. 3a).

4. On pg. 16, line 8, could you include information about how you generated your standard curve for your Bradford assay? Using BSA?

Thank you for noticing the omission to the standard curve. Yes, we used BSA for the Bradford standard curve. We included this additional information as follows:

(page 16, lines 8-9) Protein concentration was determined by Bradford assay⁵⁴ using bovine serum albumin (Sigma) for the standard curve.

We also included the reference to the Bradford quantification method:

*(page 24, lines 21-21) 54. Bradford, M. M. A rapid and sensitive method for the quantitation of microgram quantities of protein utilizing the principle of protein-dye binding. *Anal Biochem* 72, 248–54 (1976).*

5. On pg. 18, like 13, please provide more information about the “color solution” used for this assay.

Thank you for the suggestion. We modified the methods to include the composition of the two reagents used for the color solution:

(page 18, line 22-27) (...) 100 μ l of a color solution consisting of two parts of reagent A [0.37% (w/v) antipyrine and 0.25% (w/v) ammonium iron(III) sulfate in 25% (v/v) H_2SO_4 95% and 25% (v/v) H_3PO_4 85%] and one part of reagent B [0.4% (w/v) diacetylmonoxime in 7.5% (w/v) NaCl]. Reagent B is light sensitive and was stored at 4 °C in the dark. The color reagents were gently mixed and stored at 4 °C in the dark until added to the sample.

Reviewer #3 (Remarks to the Author):

In the manuscript "Mechanisms of feedback inhibition and sequential firing of active sites in plant aspartate transcarbamoylase," Bellin et al. thoroughly investigated structures of plant aspartate transcarbamoylase (ATC) in the de novo pyrimidine biosynthetic pathway using a combination of diverse experimental techniques, including in vivo plant analysis and biophysical analysis. Since the de novo pyrimidine synthesis pathway has potential value for a variety of pharmacological and agricultural applications, understanding the regulatory and catalytic mechanisms of plant ATCs is able to contribute to the research field of plant biotechnology significantly. Although the authors have proposed unique regulatory mechanisms based on multiple crystal structures and biophysical experiments, they must provide more details to support their hypotheses.

We thank Reviewer #3 for the careful revision and for the suggestions to improve the manuscript.

1. "ATC is key for plant development and photosynthetic efficiency" (Page 4, Lines 17-26 & Figure 1. 3) – The authors measured ATC transcript levels in knockdown and overexpressing lines (n = 3). More numbers of plants need to be tested.

Thank you for pointing to this shortcoming. We did three repetitions of the experiment and in each experiment 3 individual plants per line were tested. In the initial manuscript we only reported the results from one experiment (n=3). As suggested, we now include the measurements of the 9 individual plants per line. The recalculated values are consistent with the previous values and do not affect any conclusions drawn. Figure 1 and text description have been changed accordingly.

2. Crystal structure of Arabidopsis ATC bound to UMP, Table 1. – While most of the crystal structures are well processed, some of the statistics question data reliability. In particular, the I/sigma value of 6YSP (+ PALA + CP), 10.71 (1.35), is very low. Please reprocess the data set or provide other parameters of the highest resolution shell.

We revised the diffraction statistics and truncated the datasets to remove weak diffraction data. As modified in the revised Table 1, the new high-resolution cutoff for the datasets 6YPO (+UMP), 6YSP (+PALA +CP) and 6YVB (+CP) are 1.71 Å, 1.44 Å and 1.87 Å, respectively. Now, all datasets have an I/sigma(I) of at least 2.0 in the highest resolution shell.

We show below to Reviewer #3 the resolution limits, Rmerge, I/s(I), completeness and CC1/2 values for all dataset to judge the quality of the data at the highest resolution shells. Please notice that in the new Table 1, the CC1/2 is not included to comply with the journal's standard format for the crystallographic table.

	+UMP	APO	+PALA	+PALA +CP	+CP	F161A +UMP	F161A +PALA
Resolution (Å)	48.38-1.71 (1.74- 1.71)*	92.85-3.07 (3.18-3.07)	66.04-1.55 (1.60-1.55)	94.57-1.44 (1.46- 1.44)	76.16-1.87 (1.90- 1.87)	45.18-2.40 (2.49-2.40)	46.32-1.68 (1.71-1.68)
R _{merge}	0.098 (1.23)	0.14 (0.83)	0.058 (0.619)	0.085 (0.828)	0.049 (0.986)	0.194 (1.255)	0.053 (0.840)
I / σ I	14.0 (2.0)	11.2 (2.3)	15.1 (2.4)	11.1 (2.0)	18.7 (2.0)	8.5 (2.0)	18.4 (2.0)
CC1/2	0.997 (0.678)	0.994 (0.686)	0.999 (0.752)	0.995 (0.666)	0.999 (0.709)	0.992 (0.629)	0.999 (0.701)

Completeness (%)	100 (100)	99.3 (99.6)	97.4 (96.7)	100 (100)	100 (100)	100 (100)	100 (100)
------------------	-----------	-------------	-------------	-----------	-----------	-----------	-----------

The models have been refined against the re-processed datasets and the new refinement statistics are included in Table 1.

3. Page 5. Line 26: “In addition, atATC has the CP-loop (aa 156–169) and Asp-loop (aa 309–332) (Fig. 2a,d) that in other ATCs undergo large conformational movements upon substrate binding”

– The authors need to display detailed information about the large conformational movement in the figure.

We thank Reviewer #3 for the comment, since we realized that this sentence is misleading. At this stage in the Results, we do not want to show the movements of the CP- and Asp-loops in Arabidopsis ATC. We just wanted to say that the CP- and Asp-loops are present in the structure, and that in other ATCs, these loops have been shown to undergo conformational movements. The movements of the CP- and Asp-loops in Arabidopsis ATC are shown later in the manuscript, with the APO and +PALA structures. Thus, we have modified the sentence as follows:

(page 5, lines 26-28) In addition, atATC has the CP-loop (aa 156–169) and Asp-loop (aa 309–332) (Fig. 2a,d) that, as in other ATCs^{16,24}, undergo large conformational movements upon substrate binding (see below).

We believe that the movements of the CP-loop are sufficiently represented in the manuscript in Figure 3c, Figure 4a,b, Figure 5e,g and Supplementary Figure 3, and do not require an additional figure. However, the movement of the Asp-loop is only represented in Figure 3c, and to make it clearer, we have modified the figure, indicating with a red arrow the 29° rotation movement around an axis that is also represented in red. The modified figure and figure legend

are shown below.

Figure 3. (...) **c** Superposition of the three subunits in the PALA-bound trimer (colored as in **b**) and in the UMP-bound subunit (in black). The arrows indicate the closure of the subunit upon PALA binding. The Asp-loop undergoes a 29° rotation around an axis that has been represented and colored in red.

4. Page 5., Line 29: “Fig. 2d,e and Supplementary Fig. 1” – The electron density maps are provided in Supplementary Fig 2.

Thank you for noticing this mistake. It has been corrected.

5. Page 6. Lines 14-19: “atATC only binds one molecule of PALA per trimer” – The authors showed the effect of PALA [N-(phosphonacetyl)-L-aspartate] on root development. However, the number of plant samples and the error values are not indicated.

Thank you for pointing to this shortcoming. We have recalculated root length for $n=30$, and modified the text accordingly, see also response to Reviewer 2 point 3.

6. Page 6., Line 23. “Surprisingly, the structure showed the atATC trimer with PALA bound to only one of the subunits (Fig. 3b)” – As shown in Supplementary Fig 2., provide the electron density maps of 1. PALA, 2. empty, and 3. two sulfate ions and one glycerol molecule in each monomer.

We thank Reviewer #3 for the suggestion. We have made a supplementary figure (see below) showing detailed views of the electron density maps for the subunits with the active site occupied by 1. PALA, 2. two sulfate ions or 3. empty (**panel a**). We also prepared similar views of the three subunits in the atATC +PALA +CP trimer, whose active sites are occupied by 1. PALA, 2. CP, glycerol and one sulfate ion, and CP alone (**panel b**). Additionally, we show the three active sites in the atATC trimer crystallized with CP (**panel c**), and the three active sites of F161A with PALA (**panel d**).

Supplementary Figure 5. atATC active site contents. Detail of the active sites in the atATC-WT trimer crystallized with PALA (**a**), PALA and CP (**b**) and CP alone (**c**), and of the atATC-F161A trimer with PALA (**d**). The $2F_{obs}-F_{calc}$ electron density map is represented in blue mesh. Water molecules are shown as red spheres. Glycerol molecules in **b** and **c** were modelled in two alternate conformations to best explain the electron density.

Provide the B-factor value of PALA.

The average B-factor for the PALA molecule in the WT +PALA structure is 16.5 Å². We have prepared the image below for Reviewer #3, to show that the B-factors of PALA (written in red) are similar to the B-factors of the surrounding elements in the active site (written in black).

7. Page 7., Line 10, Isothermal titration calorimetry (ITC) analyses – The authors should provide all parameters of their ITC experiment results, including ΔG (kcal/mol), ΔH (kcal/mol), $-T\Delta S$ (kcal/mol), and error values. There is currently insufficient data to support the author's hypothesis because K_d values alone cannot determine the quality of the data.

We thank Reviewer #3 for the suggestion. We have changed Table 2 as suggested (see below). In addition, to show the quality of the ITC data, we have prepared a new supplementary figure with the ITC titrations (see below).

Supplementary Figure 7. ITC data. Calorimetric titrations for the interaction of atATC-WT and atATC-F161A with PALA, UMP and CP. Upper plots show the thermogram (thermal power as a function of time) and lower plots show the binding isotherm (normalized heat per injection as a function of the molar ratio ligand/protein). Non-linear fitting analysis was performed as described in Methods.

Table 2. Determination of ligand binding affinities by isothermal titration calorimetry (ITC)

	Ligand	N ^c	K _D (μM) ^a			ΔG (kcal mol ⁻¹) ^a			ΔH (kcal mol ⁻¹) ^a			-TΔS (kcal mol ⁻¹) ^a			n ^b
			Site 1	Site 2	Site 3	Site 1	Site 2	Site 3	Site 1	Site 2	Site 3	Site 1	Site 2	Site 3	
WT	PALA	1	0.59	–	–	-8.5	–	–	13.1	–	–	-21.6	–	–	1.2
	UMP	3	0.21	2.3	1.6	-9.1	-7.7	-7.9	-2.8	1.2	-0.5	-6.3	-8.9	-7.4	1.1
	CP	3	77	77	77	-5.6	-5.6	-5.6	-3.0	-3.0	3.0	-2.6	-2.6	-2.6	(1)
WT:PALA ^d	UMP	2	with PALA	1.2	1.2	with PALA	-8.1	-8.1	with PALA	0.4	0.4	with PALA	-8.5	-8.5	0.9
	CP	2	with PALA	140	140	with PALA	-5.3	-5.3	with PALA	-5.7	-5.7	with PALA	0.4	0.4	(1)
F161A	PALA	1	0.12	–	–	-9.4	–	–	11.0	–	–	-20.4	–	–	1.2
	UMP	–	–	–	–	–	–	–	–	–	–	–	–	–	–
	CP	3	0.73	0.73	0.73	-8.4	-8.4	-8.4	-6.0	-6.0	-6.0	-2.4	-2.4	-2.4	1.1

^a Relative error in dissociation constant K_D is 30%. Absolute error in binding Gibbs energy ΔG is 0.1 kcal mol⁻¹. Absolute error in binding enthalpy ΔH and entropy -TΔS is 0.4 kcal mol⁻¹.

When the binding parameters for a given interaction are equal for the three sites, ligand binding showed no cooperativity (i.e., independent binding); when the binding parameters are different, ligand binding showed cooperativity. In all cases, the binding parameters represent site-specific microscopic parameters for each binding site (i.e., intrinsic site-specific binding parameters modulated by cooperativity factors if applicable).

^b Fraction of active (ligand binding-competent) protein. Parentheses indicate the parameter n was kept fixed during the fitting analysis due to the low binding affinity.

^c Number of binding sites per protein trimer according to the ligand binding model.

^d Protein prebound to PALA in the calorimetric cell.

8. Figure 5e,g, Figure 4a,b – Use consistently defined secondary structures (e.g., helices) to compare the same regions of two structures.

We do not see inconsistencies in the representation of the secondary structures in Figures 4a,b and 5e,g. However, in figure 4d, the 3₁₀ α-helix was not represented as cartoon. Thus, we have modified panel 4d to show the helical turn (see image below) and to better compare with panels 4b and 5g.

9. Page 8., Line 16 “Mutation F161A abolishes UMP inhibition and allows binding of three PALA molecules” – Detailed kinetic analysis is required to confirm the effect of mutation F161A on the enzymatic activity. Provide steady-state kinetic parameters in a table, including enzyme efficiency, to compare WT and mutant activities.

We thank Reviewer #3 for this indication. As suggested, we have modified the text to include additional kinetic parameters:

(page X, lines Y-Z) Initial-rate plots of WT with CP as variable ligand are hyperbolic in absence of UMP ($V_{max} = 91.67 \pm 3.21 \text{ nmol min}^{-1} \mu\text{g}^{-1}$, $K_{0.5}^{CP} = 0.46 \pm 0.10 \text{ mM}$, $K_{0.5}^{Asp} = 0.94 \pm 0.26 \text{ mM}$), but turn sigmoidal in presence of UMP, with a Hill-coefficient $h = 2.2$, indicating positive cooperativity for CP binding (Fig. 5a and Supplementary Fig. 8), as previously described for wheat germ ATC⁷. In contrast, parallel assays with F161A proved that although kinetic parameter are similar to the WT ($V_{max} = 111.13 \pm 14.59 \text{ nmol min}^{-1} \mu\text{g}^{-1}$, $K_{0.5}^{CP} = 0.61 \pm 0.13 \text{ mM}$, $K_{0.5}^{Asp} = 3.87 \pm 1.36 \text{ mM}$), the activity is not inhibited by UMP and becomes more sensitive to the presence of PALA (Fig. 5a and Supplementary Fig. 8). In addition, F161A showed decreased activity at high substrate concentrations, with an inhibition constant (K_i) of 4.78 mM, whereas this substrate inhibition effect was not apparent in the WT (Fig. 5b,c and Supplementary Fig. 8).

To show this better, we prepared a supplementary figure (see below) showing the variation of the initial rate of WT and F161A with increase CP and Asp concentrations, and a side-by-side comparison of the kinetic parameters. Please notice that the WT and F161A have different inhibitory mechanisms, and their kinetics curves obey different equations. For instance, in absence of inhibitors, WT obeys the Michaelis-Menten equation [$v = V_{max} \cdot X / (K_M + X)$], whereas F161A exhibits inhibition at high substrate concentrations. Thus, the kinetic data of F161A was fitted to a modified M-M equation that corrects for such inhibitory effect and incorporates an inhibition constant (K_i).

$$v = V_{max} \cdot X / (K_{0.5} + X(1 + X/K_i)) \quad (\text{eq. 1})$$

We have explained this in the figure legend.

Supplementary Figure 8. Kinetics of atATC-WT and atATC-F161A. **a** Dependence of the initial rate (v) of atATC WT and F161A on the concentration of CP (**a**) or Asp (**b**) at fix concentration of the other substrate. The curves were fitted to a Michaelis-Menten equation with an additional term for inhibition by excess substrate: $v = V_{max} \cdot X / K_{0.5} + X(1+X/K_i)$; where X is the substrate concentration, $K_{0.5}$ is the substrate concentration at which v is one-half of the maximum velocity (V_{max}) and K_i is the inhibition constant. Best-fitting values are indicated below the graphs. atATC WT does not show inhibition by substrate since the K_i is exceedingly large and thus, obeys the Michaelis-Menten equation and $K_{0.5} = K_M$

Unfortunately, a similar side-by-side comparison of the effect of UMP and PALA on WT and F161A is not possible due to the different mechanisms of inhibition. In presence of UMP or PALA, the kinetics of both enzymes are sigmoidal. But, whereas the WT curves obey the Hill equation [$v = V_{max} \cdot X^h / K_{0.5}^h + X^h$], the fitting of the F161A data needs a modification of the Hill equation to account for the effect of substrate inhibition:

$$v = V_{max} \cdot X^h / K_{0.5}^h + (X^h (1 + X/K_i)) \quad (\text{eq. 2})$$

Equations 1 and 2 fit well the WT and F161A data and the difference between them is clearly shown in Figure 5. However, it is difficult to compare the kinetic parameters since the best-fitting values (V_{max} , $K_{0.5}$, K_i and h) of eq. 2 do not unambiguously define the kinetic curves of F161A. This is, many combinations of these parameters generate curves that fit the data equally well. We have modified the Activity assays subheading in the Methods section to report the equations used in the kinetic analysis and to explain the difficulties encountered with the fitting of the F161A sigmoidal curves.

(page 18, line 30 - page 19, line 9) Data analysis was done with GraphPad. In absence of inhibitors, WT kinetics obey Michaelis-Menten equation $[V_{max} \cdot X / K_{0.5} + X(1+X)]$, where X is substrate concentration and $K_{0.5}$ is the substrate concentration at which the initial rate (v) is one-half of the maximum velocity (V_{max}). In absence of UMP or PALA, F161A shows inhibition by excess of substrate and the data was fitted to equation $v = V_{max} \cdot X / K_{0.5} + X(1+X/K_i)$, where K_i is the inhibition constant. In presence of UMP or PALA, WT kinetics obey the Hill equation $[v = V_{max} \cdot X^h / K_{0.5}^h + X^h]$, whereas F161A kinetics were fitted to a modified equation with the additional term for substrate inhibition: $v = V_{max} \cdot X^h / K_{0.5}^h + (X^h(1+X/K_i))$. Best-fitting values for the latter equation are ambiguous, since many combinations of these parameters generate curves that fit the data equally well.

Reviewer #3 suggests to include the “enzyme efficiency” to compare WT and F161A activities. Although the k_{cat}/K_M ratio is often used to compare the specificity of an enzyme for different substrates, we would prefer not to use this value as an indicator of the catalytic efficiency, as has been defended by other authors (Eisenthal R, Danson MJ, Hough DW. *Catalytic efficiency and k_{cat}/K_M : a useful comparator?* Trends Biotechnol. 2007 25(6):247-9). Thus, although k_{cat}/K_M ratios are not reported for WT and F161A, the reader could do such comparison easily from the V_{max} and $K_{0.5}$ values reported in Supplementary Figure 8.

10. Page 9., Lines 9-11. “Although ITC indicated that PALA binds with high affinity to only one site per trimer (Table 2), the high concentration of PALA (2 mM) in the crystallization condition must favor a low-affinity binding to the other subunits.” – Given the low binding affinity in Table2, it is difficult to explain the three ligand bindings in the crystal structure. In other words, if the three PALA bindings can be seen at a concentration of 2mM, the authors could be able to calculate the K_d value in the ITC experiment.

We shared the Reviewer’s concerns, and found a number of reasons for the apparent discrepancy between the crystallization and ITC results:

1. Protein concentration for crystallization was 5 mg ml^{-1} (135 μM), whereas for ITC experiments were done at 30-40 μM protein. Since binding of PALA is an equilibrium, a higher protein concentration would displace the equilibrium towards the formation of the complex with PALA in the crystal.
2. The concentration of PALA in the crystallization condition is 2 mM, whereas in ITC, the syringe is loaded with 0.4 mM PALA and makes 19 injections of 2 μl into a calorimetric cell with 200 μl of sample. Thus, by the end of the ITC experiment, the maximum concentration of PALA in the cell reached 0.07 mM, 28-fold lower than the concentration of PALA in the crystallization condition. As before, a higher concentration of PALA would favor the displacement of the equilibrium towards the formation of the complex in the crystal.
3. ITC detects the heat absorbed or produced by the initial encounter between protein and ligand as a function of time (transient effect during the first minute after mixing). If binding of PALA to the other binding sites were a slow process, the heat associated to that binding would be dissipated in the 1-2 hours that last the experiment time (or even at longer times) and the corresponding thermal power would not be distinguishable from the baseline. In contrast, crystallization is a process that lasts weeks and thus, could possibly favor the slow binding of PALA to the low-affinity subunits.
4. The crystallization condition contains a high % of PEG that decreases the availability of solvent, increasing the concentration of protein and ligand, thus favoring the displacement of the equilibrium towards the formation of the complex.
5. We cannot discard that the packing of the protein in the crystal lattice could favor the closed conformation with higher affinity for PALA.

Editorial comments

Following Nature Communication formatting instructions and editorial policy we made the following changes:

- Results and Methods subheadings were shortened
- Subheadings from Discussion have been removed.
- Crystallographic table was modified to comply with journal's standard format.
- We added Data Availability, Author Contributions and Competing Interests sections.
- To better show the data distribution, bar graphs in Figure 1c and 1e now show the scatter dot plot. Similarly, Figure 5a-c and Supplementary Figure 8 have been modified to represent all data points instead of the mean and SD.

REVIEWER COMMENTS

Reviewer #1 (Remarks to the Author):

no further comments

Reviewer #2 (Remarks to the Author):

Thank you for your detailed responses. The authors have sufficiently addressed all of my concerns.

Reviewer #3 (Remarks to the Author):

In the revised manuscript “Mechanisms of feedback inhibition and sequential firing of active sites in plant aspartate transcarbamoylase,” Bellin et al. well addressed the reviewers’ comments on the X-ray crystallography part. Compared to the previous manuscript, structural analysis is more consistent, and the revised statistics of X-ray diffraction data are more reliable. However, the authors did not adequately address the problems of Isothermal titration calorimetry (ITC) analysis and enzyme kinetic analysis.

[ITC Analysis] Table 2.

 As suggested, the authors provide more experimental results, including ΔG (kcal/mol), ΔH (kcal/mol), $-T\Delta S$ (kcal/mol). However, to determine the quality of the ITC experiments and data-fitting, the authors should provide the error values for each fitting. In particular, please provide the error values of CP / WT, UMP / WT:PALA), and CP / WT:PALA.

[ITC Analysis] Supplementary Figure 7.

 In the revised manuscript, the authors provide titration curves for the interactions of WT and F161A with PALA, UMP, and CP. As mentioned above, the titration curves, especially Supplementary Fig 7(c) (i.e., CP / WT), 7(d) (i.e., UMP / WT:PALA), and 7(e) (CP / WT:PALA), raise questions about data quality and data-fitting. The authors should use a broader range (higher) concentrations or

adjust the cell-syringe concentrations to improve data quality. Since the current titration curves for S7 (c), (d), and (e) are only part of the overall titration, there are problems with the reliability of the data. Because the ITC data (Table 2. and Supplementary Figure 7.) is one of the critical supporting data for the author's hypothesis, it is essential to prove the reliability of the data or provide parameters that can be used to determine the data quality.

[ITC Analysis] Page 9, lines 202 – 204, “and also shows 100-fold higher affinity for CP (KDCP=0.7 μ M) (Table 2 and Supplementary Fig. 7).”

&

Page 11, lines 263 – 265 “atATC has a surprising affinity for UMP, \sim 400-fold higher than for CP (Table 2), which is explained by the extensive contacts of the nucleotide with the CP and Asp binding sites, similar to what PALA does.”

 For example, these quantitative comparisons cannot be used in this manuscript without the ITC data's reliability.

[ITC Analysis] Supplementary Figure 7. & Page 9, line 200 “ITC analysis failed to detect the binding of UMP to F161A”

 Please provide the titration curve for F161A / UMP in Supplementary Figure 7, although it failed to detect the binding.

[ITC Analysis & X-ray Crystallography] Supplementary Figure 7. & Page 8, line 187 “Mutant F161A is not inhibited by UMP and binds three PALAs”

 The heat signals of F161A / PALA (Supplementary Figure 7. (f)) seem to be sufficient for in-depth analysis of the binding stoichiometry. For the conclusion, which is “F161A binds three PALAs,” authors have to explain more about the discrepancy between the structural and ITC analyses in the main manuscript. Otherwise, the authors cannot exclude the possibility of crystallographic artifacts.

[Enzyme Kinetics] Fig 5. & Supplemental Figure 8.

 Although authors explain why they do not provide “enzyme efficiency” in this author's response, the k_{cat}/K_m value is the most commonly used parameter to compare the enzymatic efficiency of WT and mutations. Based on the approximate calculations using the kinetic parameters provided, the enzymatic efficiencies of WT and F161A are nearly identical. I believe many other Nature Communications readers also will ask the same question. The authors need to explain this as well.

[Enzyme Kinetics] Fig 5. & Supplemental Figure 8. & Fig 5., Page 9, lines 197-199, "In addition, F161A showed decreased activity at high substrate concentrations, with an inhibition constant (K_i) of 4.78 mM, whereas this substrate inhibition effect was not apparent in the WT"

 The data points at the highest concentrations (e.g., 2.5 mM CP and 10 mM ASP) are highly variable. As the authors analyzed, it probably could be the substrate inhibition effect. However, if possible, authors repeat the enzyme assays and provide better data sets.

Reviewer #1:

no further comments

Reviewer #2:

Thank you for your detailed responses. The authors have sufficiently addressed all of my concerns.

We thank Reviewers #1 and #2 for the revision of the manuscript.

Reviewer #3:

In the revised manuscript "Mechanisms of feedback inhibition and sequential firing of active sites in plant aspartate transcarbamoylase," Bellin et al. well addressed the reviewers' comments on the X-ray crystallography part. Compared to the previous manuscript, structural analysis is more consistent, and the revised statistics of X-ray diffraction data are more reliable. However, the authors did not adequately address the problems of Isothermal titration calorimetry (ITC) analysis and enzyme kinetic analysis.

We thank Reviewer #3 for the careful revision of the manuscript and for helping us to make the data appear more consistent and reliable.

1. ITC Analysis - Table 2. As suggested, the authors provide more experimental results, including ΔG (kcal/mol), ΔH (kcal/mol), $-T\Delta S$ (kcal/mol). However, to determine the quality of the ITC experiments and data-fitting, the authors should provide the error values for each fitting. In particular, please provide the error values of CP / WT, UMP / WT:PALA), and CP / WT:PALA.

As suggested, we have modified Table 2 (page 32) to show explicitly the errors for each estimated value. We hope that this will help in observing significant differences in the parameters corresponding to the different interactions.

2. ITC Analysis - Supplementary Figure 7. In the revised manuscript, the authors provide titration curves for the interactions of WT and F161A with PALA, UMP, and CP. As mentioned above, the titration curves, especially Supplementary Fig 7(c) (i.e., CP / WT), 7(d) (i.e., UMP / WT:PALA), and 7(e) (CP / WT:PALA), raise questions about data quality and data-fitting. The authors should use a broader range (higher) concentrations or adjust the cell-syringe concentrations to improve data quality. Since the current titration curves for S7 (c), (d), and (e) are only part of the overall titration, there are problems with the reliability of the data. Because the ITC data (Table 2. and Supplementary Figure 7.) is one of the critical supporting data for the author's hypothesis, it is essential to prove the reliability of the data or provide parameters that can be used to determine the data quality.

The CP/WT titration is characterized by $c=0.17$, the UMP/WT:PALA titration by $c=12$, and the CP/WT:PALA by $c=0.1$. Due to the low affinity for CP/WT and CP/WT:PALA interactions, even raising the concentration in the syringe would not result in a practical improvement in the shape of the titration, and problems with the dilution of highly concentrated ligands in the syringe would arise (large background injection heat). For significantly improving the c parameter, the cell concentration should be raised to impractical values (at least a factor of 10), and the same factor or larger for the syringe concentration. Performing experiments at substantially different protein concentrations in the cell would introduce additional factors into play and additional effects may appear. According to Tellinghuisen (*Tellinghuisen J. Isothermal titration calorimetry at very low c. Anal Biochem. 2008, 373(2):395-7*), at low c values it is possible to reliably estimate the binding affinity even if a considerable uncertainty affects the enthalpy and the stoichiometry; therefore, increasing the ligand concentration is not going to reduce much more the uncertainty in binding affinity. We believe that, within the errors indicated in Table 2, it is possible to extract reasonable conclusions about the differences in affinities.

3. ITC Analysis. Page 9, lines 202 – 204: "and also shows 100-fold higher affinity for CP (KDCP=0.7 μ M) (Table 2 and Supplementary Fig. 7). Page 11, lines 263 – 265: "atATC has a surprising affinity for UMP, ~400-fold higher than for CP (Table 2), which is explained by the extensive contacts of the nucleotide with the CP and Asp binding sites, similar to what PALA

does.” For example, these quantitative comparisons cannot be used in this manuscript without the ITC data's reliability.

As previously mentioned, we think that, with the errors indicated in Table 2, it is possible to extract reasonable conclusions about differences in affinities. Therefore, we would rather maintain those sentences indicating the approximate ~100-fold and ~400-fold differences in affinities.

4. ITC Analysis, Supplementary Figure 7. & Page 9, line 200: “ITC analysis failed to detect the binding of UMP to F161A”. Please provide the titration curve for F161A / UMP in Supplementary Figure 7, although it failed to detect the binding.

Thank you for the suggestion. The UMP/F161A titration has been included in Supplementary Figure 7 as panel h (see below), showing that no interaction was observed.

5. ITC Analysis & X-ray Crystallography - Supplementary Figure 7. Page 8, line 187: “Mutant F161A is not inhibited by UMP and binds three PALAs”. The heat signals of F161A / PALA (Supplementary Figure 7. (f)) seem to be sufficient for in-depth analysis of the binding stoichiometry. For the conclusion, which is “F161A binds three PALAs,” authors have to explain more about the discrepancy between the structural and ITC analyses in the main manuscript. Otherwise, the authors cannot exclude the possibility of crystallographic artifacts.

As mentioned in the previous response letter, we believe that the apparent discrepancy between the ITC and crystallization experiments is due to the higher concentrations of PALA (28-fold higher), UMP (70-fold higher) and protein (4.5-fold higher) used in the latter. These concentrations refer to the beginning of the crystallization experiment, since once the crystallization drop is equilibrated with the reservoir, the concentrations of protein, ligand and precipitant will be even higher. Following the suggestion, we have modified the text to be more explicit about these differences:

(page 9, lines 15–25) In apparent contradiction with the activity and ITC results, the structure showed a molecule of UMP in the active site (Fig. 5d), the CP-loop in the inhibited conformation, and the missing F161 side chain being replaced by a glycerol molecule (Fig. 5e). Likely, the low-affinity binding of the nucleotide to the active site of the mutated protein is favored by the higher concentrations of nucleotide (5 mM) and protein (135 µM) used in the crystallization condition compared to those in the ITC experiments (70 µM UMP at most and 30-40 µM protein). We also determined the structure of F161A crystallized with PALA (Table 1). Interestingly, the structure showed a trimer bound to three molecules of PALA rather than one as in the WT (Fig. 5f and Supplementary Fig. 5). Although ITC indicated that PALA binds with high affinity to only one site per trimer (Table 2), the high concentrations of PALA (2 mM)

and protein in the crystallization condition must favor a low-affinity binding to the other subunits.

6. Enzyme Kinetics Fig 5. & Supplemental Figure 8. Although authors explain why they do not provide “enzyme efficiency” in this author's response, the k_{cat}/K_M value is the most commonly used parameter to compare the enzymatic efficiency of WT and mutations. Based on the approximate calculations using the kinetic parameters provided, the enzymatic efficiencies of WT and F161A are nearly identical. I believe many other Nature Communications readers also will ask the same question. The authors need to explain this as well.

We have included in Supplementary Figure 8 the k_{cat} for the WT and F161A proteins, and as suggested by Reviewer #3, we have also added the $k_{cat}/K_{0.5}$ value for both enzymes (see below). Please notice that we refer to $K_{0.5}$ instead of K_M because F161A does not obey the Michaelis-Menten equation.

	WT	F161A
Best-fit values		
V_{max}	93.23	115.0
$K_{0.5}^{CP}$	0.46	0.62
K_i	>>>	4.52
Std. error		
V_{max}	2.53	14.62
$K_{0.5}^{CP}$	0.04	0.13
K_i	-	1.64
95% confidence interval		
V_{max}	88.07 - 98.38	83.83 - 146.2
$K_{0.5}^{CP}$	0.39 - 0.54	0.35 - 0.89
K_i	-	1.03 - 8.01
k_{cat} (min ⁻¹)	3,463	4,273
$k_{cat}/K_{0.5}^{CP}$ (min ⁻¹ mM ⁻¹)	7,530	6,891

	WT	F161A
Best-fit values		
V_{max}	85.96	128.4
$K_{0.5}^{Asp}$	0.94	3.20
K_i	>>>	16.08
Std. error		
V_{max}	2.51	26.71
$K_{0.5}^{Asp}$	0.10	1.03
K_i	-	8.51
95% confidence interval		
V_{max}	80.80 - 91.13	71.52 - 185.4
$K_{0.5}^{Asp}$	0.73 - 1.16	1.00 - 5.40
K_i	-	0 - 34.23
k_{cat} (min ⁻¹)	3,194	4,771
$k_{cat}/K_{0.5}^{Asp}$ (min ⁻¹ mM ⁻¹)	3,398	1,491

As correctly pointed by the reviewer, the proficiency of both enzymes in catalyzing the reaction is similar, except for the fact that F161A is not inhibited by UMP, is more sensitive to the inhibition

by PALA and shows decreased activity at high substrate concentrations. As suggested, we have modified the text to better explain this point:

(page 8, lines 24–26) “The mutation did not affect the solubility nor the oligomeric state of the protein, but changed the susceptibility of the enzyme to UMP, PALA or to high concentrations of the substrates.”

(page 8, line30 – page 9, line 6) “In contrast, parallel assays with F161A proved that although the catalytic activity is highly similar to the WT (...), the enzyme is not inhibited by UMP and also becomes more sensitive to the presence of PALA (Fig. 5a and Supplementary Fig. 8). In addition, F161A showed decreased activity at high substrate concentrations, with an inhibition constant (K_i) of 4.52 mM, whereas this substrate inhibition effect was not apparent in the WT (Fig. 5b,c and Supplementary Fig. 8).”

(page 11, lines 1–3) “Indeed, a single point mutation in the CP-loop, F161A, is sufficient to turn atATC insensitive to UMP without affecting the catalytic efficiency of the enzyme nor its inhibition by PALA”

However, we do not want to explicitly refer to the k_{cat}/K_M value to compare both enzymes. Although included in Supplementary figure 8, we want to express our disconformity in using the k_{cat}/K_M to compare the catalytic activities, particularly, when the two enzymes show different inhibitory mechanisms and kinetic behaviors. We hope that the reviewer will respect our position, which is defended by experts in the field:

Athel Cornish-Bowden. Fundamentals of Enzyme Kinetics. 4th Edition. Wiley-Blackwell. p. 43:

“...there has been a regrettable tendency to use the k_{cat}/K_M , for comparing different enzymes as catalysts, such as mutant forms of an enzyme obtained by genetic manipulation, and not just for comparing different substrates for the same enzyme. Eisenthal and co-workers have pointed out [R Eisenthal, MJ Danson, DW Hough (2007) “Catalytic efficiency and k_{cat}/K_M : a useful comparator? Trends in Biotechnology 25, 247-249] that this can sometimes lead to incorrect conclusions, because enzymes with the same value of k_{cat}/K_M and acting on the same substrates may have different rate ratios at different substrate concentrations, and the enzyme with the higher value of k_{cat}/K_M may still give the lower rate in some concentration range. Put simply, the enzyme with the higher value of k_{cat}/K_M may will give the faster reaction at low substrate concentrations, whereas the enzyme with the higher value of k_{cat} will give the faster reaction at high substrate concentrations: as there is no necessity for these two conditions to agree, there is no necessity for the enzyme with the higher value of k_{cat}/K_M to give the faster reaction over the whole concentration range. It follows, therefore, that its use as a measure of specificity should not be generalized beyond the original idea of comparing the kinetics of different substrates for the same enzyme.”

7. Enzyme Kinetics - Fig 5. & Supplemental Figure 8. & Fig 5., Page 9, lines 197-199, “In addition, F161A showed decreased activity at high substrate concentrations, with an inhibition constant (K_i) of 4.78 mM, whereas this substrate inhibition effect was not apparent in the WT” . The data points at the highest concentrations (e.g., 2.5 mM CP and 10 mM ASP) are highly variable. As the authors analyzed, it probably could be the substrate inhibition effect. However, if possible, authors repeat the enzyme assays and provide better data sets.

We have performed many kinetic assays to convince ourselves beyond doubt that the activity of F161A decreases at high concentrations of both substrates. As shown in figure 5b, we observe this behavior both in the absence and in the presence of UMP or PALA. These enzymatic assays are quite laborious. Each point plotted in the graphs is the result of measuring the initial rate in a time-course experiment with at least 5 measurements over time. Obtaining results of this quality in this assay is a major effort. Nonetheless, we agree with the reviewer that the points at higher concentration showed higher variability. Following the advice, we repeated the assays for the F161A mutant in absence of inhibitors. The new results are similar to previous experiments and fit the kinetic equation with the substrate inhibition term. The new curves and kinetic parameters are shown in Figure 5 and Supplementary Figure 8 (see figure above).

Editor and Reviewers: Additional corrections

We recently observed variations in the quantification of the ATC protein contents in plant extracts. This variability likely depends on small differences during the harvesting and extraction procedures. Being aware of this, we repeated the Western blot experiments shown in Figure 1d to have a more reliable comparison of the ATC levels in Col-0 versus down- and upregulated plants. The new calculated values do not affect the conclusions drawn in our work and are included in the text as follows:

(page 4, lines 19-23) Two selected lines, atc-1 and atc-2, exhibited 16% and 10% residual ATC transcript and a 3-fold or 20-fold drop in protein levels compared to wild-type (WT; Col-0) controls (Fig. 1b–d). Conversely, we constitutively overexpressed ATC in two Arabidopsis lines, ATC-OX1 and ATC-OX2, which showed 13- and 16-fold increase in ATC transcript and a 2.9-fold increase in protein levels (Fig. 1b–d).

(Figure 1 legend, page 27, lines 13-15) d Immunoblot with anti-ATC antibody on whole leaf extracts; Coomassie Brilliant Blue (CBB) stained SDS-PAGE was used as loading control. The ATC protein levels relative to Col-0 quantified from n-different experiments are: atc-1, 0.34 ± 0.17 (n=5); atc-2, 0.05 ± 0.024 (n=5); ATC-Ox1, 2.9 ± 0.44 (n=4); ATC-Ox2, 2.9 ± 0.60 (n=4).

The new blots together with the quantification results and loading controls are now included in the raw data file.

REVIEWERS' COMMENTS

Reviewer #3 (Remarks to the Author):

The authors well addressed the reviewers' comments. While concerns remain about the discrepancy between crystallography and ITC data, the updated table contains the data that readers need to determine their ITC experiments' quality. If possible, I recommend including the author's response to comment #2 in the discussion, but this is not required.